# A Regularized Approach to Sparse Optimal Policy in Reinforcement Learning

**Xiang Li**[*]
School of Mathematical Sciences
Peking University
Beijing, China
`lx10077@pku.edu.cn`

**Wenhao Yang**[*]
Center for Data Science
Peking University
Beijing, China
`yangwenhaosms@pku.edu.cn`

**Zhihua Zhang**
National Engineering Lab for Big Data Analysis and Applications
School of Mathematical Sciences
Peking University
Beijing, China
`zhzhang@math.pku.edu.cn`

## Abstract

We propose and study a general framework for regularized Markov decision processes (MDPs) where the goal is to find an optimal policy that maximizes the expected discounted total reward plus a policy regularization term. The extant entropy-regularized MDPs can be cast into our framework. Moreover, under our framework, many regularization terms can bring multi-modality and sparsity, which are potentially useful in reinforcement learning. In particular, we present sufficient and necessary conditions that induce a sparse optimal policy. We also conduct a full mathematical analysis of the proposed regularized MDPs, including the optimality condition, performance error, and sparseness control. We provide a generic method to devise regularization forms and propose off-policy actor critic algorithms in complex environment settings. We empirically analyze the numerical properties of optimal policies and compare the performance of different sparse regularization forms in discrete and continuous environments.

## 1 Introduction

Reinforcement learning (RL) aims to find an optimal policy that maximizes the expected discounted total reward in an MDP [4, 36]. It's not an easy task to solve the *nonlinear* Bellman equation [2] greedily in a high-dimension action space or when function approximation (such as neural networks) is used. Even if the optimal policy is obtained precisely, it is often the case the optimal policy is deterministic. Deterministic policies are bad to cope with unexpected situations when its suggested action is suddenly unavailable or forbidden. By contrast, a multi-modal policy assigns positive probability mass to both optimal and near optimal actions and hence has multiple alternatives to handle this case. For example, an autonomous vehicle aims to do path planning with a pair of departure and destination as the state. When a suggested routine is unfortunately congested, an alternative routine could be provided by a multi-modal policy, which can't be provided by a deterministic policy without evoking a new computation. Therefore, in a real-life application, we hope the optimal policy to possess thee property of multi-modality.

---

[*]Equal contribution.

Entropy-regularized RL methods have been proposed to handle the issue. More specifically, an entropy bonus term is added to the expected long-term returns. As a result, it not only softens the non-linearity of the original Bellman equation but also forces the optimal policy to be stochastic, which is desirable in problems where dealing with unexpected situations is crucial. In prior work, the *Shannon entropy* is usually used. The optimal policy is of the form of *softmax*, which has been shown can encourage exploration [8, 40]. However, a softmax policy assigns a non-negligible probability mass to all actions, including those really terrible and dismissible ones, which may result in an unsafe policy. For RL problems with high dimensional action spaces, a sparse distribution is preferred in modeling a policy function, because it implicitly does *action filtration*, i.e., weeds out suboptimal actions and maintains near optimal actions. Thus, Lee et al. [19] proposed to use *Tsallis entropy* [39] instead, giving rise to a *sparse* MDP where only few actions have non-zero probability at each state in the optimal policy. Lee et al. [20] empirically showed that general Tsallis entropy[2] also leads to a sparse MDP. Moreover, the Tsallis regularized RL has a lower performance error, i.e., the optimal value of the Tsallis regularized RL is closer to the original optimal value than that of the Shannon regularized RL.

The above discussions manifest that an entropy regularization characterizes the solution to the corresponding regularized RL. From Neu et al. [28], any entropy-regularized MDP can be viewed as a regularized convex optimization problem where the entropy serves as the regularizer and the decision variable is a stationary policy. Geist et al. [10] proposed a framework in which the MDP is regularized by a general strongly concave function. It analyzes some variants of classic algorithms under that framework but does not provide insight into the choice of regularizers. On the other hand, a sparse optimal policy distribution is more favored in large action space RL problems. Prior work Lee et al. [19], Nachum et al. [27] obtains a sparse optimal policy by the Tsallis entropy regularization. Considering the diversity and generality of regularization forms in convex optimization, it is natural to ask whether other regularizations can lead to sparseness. The answer is that there does exist other regularizers that induces sparsity.

In this paper, we propose a framework for regularized MDPs, where a general form of regularizers is imposed on the expected discounted total reward. This framework includes the entropy regularized MDP as a special case, implying certain regularizers can induce sparseness. We first give the optimality condition in regularized MDPs under the framework and then give necessary and sufficient conditions to show which kind of regularization can lead to a sparse optimal policy. Interestingly, there are lots of regularization that can lead to the sparseness, and the degree of sparseness can be controlled by the regularization coefficient. Furthermore, we show that regularized MDPs have a regularization-dependent performance error caused by the regularization term, which could guide us to choose an effective regularization when it comes to dealing with problems with a continuous action space. To solve regularized MDPs, we employ the idea of generalized policy iteration and propose an off-policy actor-critic algorithm to figure out the performance of different regularizers.

## 2 Notation and preliminaries

**Markov Decision Processes** In reinforcement learning (RL) problems, the agent's interaction with the environment is often modeled as an Markov decision process (MDP). An MDP is defined by a tuple $(\mathcal{S}, \mathcal{A}, \mathbb{P}, r, \mathbb{P}_0, \gamma)$, where $\mathcal{S}$ is the state space and $\mathcal{A}$ the action space with $|\mathcal{A}|$ actions. We use $\Delta_{\mathcal{X}}$ to denote the simplex on any set $\mathcal{X}$, which is defined as the set of distributions over $\mathcal{X}$, i.e., $\Delta_{\mathcal{X}} = \{P : \sum_{x \in \mathcal{X}} P(x) = 1, P(x) \geq 0\}$. The vertex set of $\Delta_{\mathcal{X}}$ is defined as $V_{\mathcal{X}} = \{P \in \Delta_{\mathcal{X}} : \exists\, x \in \mathcal{X}, \text{s.t. } P(x) = 1\}$. $\mathbb{P} : \mathcal{S} \times \mathcal{A} \to \Delta_{\mathcal{S}}$ is the unknown state transition probability distribution and $r : \mathcal{S} \times \mathcal{A} \to [0, R_{\max}]$ is the bounded reward on each transition. $\mathbb{P}_0$ is the distribution of initial state and $\gamma \in [0, 1)$ is the discount factor.

**Optimality Condition of MDP**

The goal of RL is to find a stationary policy which maps from state space to a simplex over the actions $\pi : \mathcal{S} \to \Delta_{\mathcal{A}}$ that maximizes the expected discounted total reward, i.e.,

$$\max_{\pi} \mathbb{E}\left[\sum_{t=0}^{\infty} \gamma^t r(s_t, a_t) \Big| \pi, \mathbb{P}_0\right],  \tag{1}$$

where $s_0 \sim \mathbb{P}_0, a_t \sim \pi(\cdot|s_t)$, and $s_{t+1} \sim \mathbb{P}(\cdot|s_t, a_t)$. Given any policy $\pi$, its state value and Q-value functions are defined respectively as

$$V^\pi(s) = \mathbb{E}\left[\sum_{t=0}^\infty \gamma^t r(s_t, a_t)|s_0 = s, \pi\right],$$

$$Q^\pi(s, a) = \mathbb{E}_{a\sim\pi(\cdot|s)}\left[r(s, a) + \gamma\mathbb{E}_{s'|s,a}V^\pi(s')\right].$$

Any solution of the problem (1) is called an *optimal* policy and denoted by $\pi^*$. Optimal policies may not be unique in an MDP, but the optimal state value is unique (denoted $V^*$). Actually, $V^*$ is the unique fixed point of the Bellman operator $\mathcal{T}$, i.e., $V^*(s) = \mathcal{T}V^*(s)$ and

$$\mathcal{T}V(s) \triangleq \max_\pi \mathbb{E}_{a\sim\pi(\cdot|s)}\left[r(s, a) + \gamma\mathbb{E}_{s'|s,a}V(s')\right].$$

$\pi^*$ often is a *deterministic* policy which puts all probability mass on one action[31]. Actually, it can be obtained as the greedy action w.r.t. the optimal Q-value function, i.e., $\pi^*(s) \in \mathrm{argmax}_a Q^*(s, a)$ [3]. The optimal Q-value can be obtained from the state value $V^*(s)$ by definition.

As a summary, any optimal policy $\pi^*$ and its optimal state value $V^*$ and Q-value $Q^*$ satisfy the following *optimality condition* for all states and actions,

$$Q^*(s, a) = r(s, a) + \gamma\mathbb{E}_{s'|s,a}V^*(s),$$
$$V^*(s) = \max_a Q^*(s, a), \ \pi^*(s) \in \mathrm{argmax}_a Q^*(s, a).$$

## 3 Regularized MDPs

To obtain a sparse but multi-modal optimal policy, we impose a general regularization term to the objective (1) and solve the following *regularized* MDP problem

$$\max_\pi \mathbb{E}\left[\sum_{t=0}^\infty \gamma^t(r(s_t, a_t) + \lambda\phi(\pi(a_t|s_t)))\Big|\pi, \mathbb{P}_0\right], \quad (2)$$

where $\phi(\cdot)$ is a regularization function. Problem (2) can be seen as a RL problem in which the reward function is the sum of the original reward function $r(s, a)$ and a term $\phi(\pi(a|s))$ that provides regularization. If we take expectation to the regularization term $\phi(\pi(a|s))$, it can be found that the quantity

$$H_\phi(\pi) = \mathbb{E}_{a\sim\pi(\cdot|s)}\phi(\pi(a|s)), \quad (3)$$

is entropy-like but not necessarily an entropy in our work. However, Problem (2) is not well-defined since arbitrary regularizers would be more of a hindrance than a help. In the following, we make some assumptions about $\phi(\cdot)$.

### 3.1 Assumption for regularizers

A regularizer $\phi(\cdot)$ characterizes solutions to Problem (2). In order to make Problems (2) analyzable, a basic assumption (Assumption 1) is prerequisite. Explanation and examples will be provided to show that such an assumption is reasonable and not strict.

**Assumption 1** *The regularizer $\phi(x)$ is assumed to satisfy the following conditions on the interval $(0, 1]$: (1) **Monotonicity**: $\phi(x)$ is non-increasing; (2) **Non-negativity**: $\phi(1) = 0$; (3) **Differentiability**: $\phi(x)$ is differentiable; (4) **Expected Concavity**: $x\phi(x)$ is strictly concave.*

The assumptions of monotonicity and non-negativity make the regularizer be an positive exploration bonus. The bonus for choosing an action of high probability is less than that of choosing an action of low probability. When the policy becomes deterministic, the bonus is forced to zero. The assumption of differentiability facilitates theoretic analysis and benefits practical implementation due to the widely used automatic derivation in deep learning platforms. The last assumption of expected concavity makes $H_\phi(\pi)$ a concave function w.r.t. $\pi$. Thus any solution to Eqn.(2) hardly lies in the vertex set of

the action simplex $V_{\mathcal{A}}$ (where the policy is deterministic). As a byproduct, let $f_\phi(x) = x\phi(x)$. Then its derivative $f'_\phi(x) = \phi(x) + x\phi'(x)$ is a strictly decreasing function on $(0, 1)$ and thus $(f'_\phi)^{-1}(x)$ exists. For simplicity, we denote $g_\phi(x) = (f'_\phi)^{-1}(x)$.

There are plenty of options for the regularizer $\phi(\cdot)$ that satisfy Assumption 1. First, entropy can be recovered by $H_\phi(\pi)$ with specific $\phi(\cdot)$. For example, when $\phi(x) = -\log x$, the Shannon entropy is recovered; when $\phi(x) = \frac{k}{q-1}(1 - x^{q-1})$ with $k > 0$, the Tsallis entropy is recovered. Second, there are many instances that are not viewed as an entropy but can serve as a regularizer. We find two families of such functions, namely, the exponential function family $q - x^k q^x$ with $k \geq 0, q \geq 1$ and the trigonometric function family $\cos(\theta x) - \cos(\theta)$ and $\sin(\theta) - \sin(\theta x)$ both with hyper-parameter $\theta \in (0, \frac{\pi}{2}]$. Since these functions are simple, we term them *basic* functions.

Apart from the basic functions mentioned earlier, we come up with a generic method to combine different basic functions. Let $\mathcal{F}$ be the set of all functions satisfying Assumption 1. By Proposition 1, the operations of positive addition and minimum can preserve the properties shared among $\mathcal{F}$. Therefore, the finite-time application of such operations still leads to an available regularizer.

**Proposition 1** *Given $\phi_1, \phi_2 \in \mathcal{F}$, we have $\alpha\phi_1 + \beta\phi_2 \in \mathcal{F}$ for all $\alpha, \beta \geq 0$ and $\min\{\phi_1, \phi_2\} \in \mathcal{F}$.*

Here we only consider those differentiable $\min\{\phi_1, \phi_2\}$ in theoretical analysis, because the minimum of any two functions in $\mathcal{F}$ may be non-differentiable on some points. For instance, given any $q > 1$, the minimum of $-\log(x)$ and $q(1 - x)$ has a unique non-differentiable point on $(0, 1)$.

### 3.2 Optimality and sparsity

Once the regularizer $\phi(\cdot)$ is given, similar to non-regularized case, the (regularized) state value and Q-value functions of any given policy $\pi$ in a regularized MDP are defined as

$$V_\lambda^\pi(s) = \mathbb{E}\Big[\sum_{t=0}^{+\infty} \gamma^t(r(s_t, a_t) + \lambda\phi(\pi(a_t|s_t)))\Big|s_0 = s, \pi\Big],$$

$$Q_\lambda^\pi(s, a) = r(s, a) + \gamma\mathbb{E}_{a \sim \pi(\cdot|s)}\mathbb{E}_{s'|s,a}V_\lambda^\pi(s'). \tag{4}$$

Any solution to Problem (2) is call the *regularized optimal* policy (denoted $\pi_\lambda^*$). The corresponding *regularized optimal* state value and Q-value are also optimal and denoted by $V_\lambda^*$ and $Q_\lambda^*$ respectively. If the context is clear, we will omit the word *regularized* for simplicity. In this part, we aim to show the optimality condition for the regularized MDPs (Theorem 1). The proof of Theorem 1 is given in Appendix B.

**Theorem 1** *Any optimal policy $\pi_\lambda^*$ and its optimal state value $V_\lambda^*$ and Q-value $Q_\lambda^*$ satisfy the following optimality condition for all states and actions:*

$$Q_\lambda^*(s, a) = r(s, a) + \gamma\mathbb{E}_{s'|s,a}V_\lambda^*(s'),$$

$$\pi_\lambda^*(a|s) = \max\left\{g_\phi\left(\frac{\mu_\lambda^*(s) - Q_\lambda^*(s, a)}{\lambda}\right), 0\right\}, \tag{5}$$

$$V_\lambda^*(s) = \mu_\lambda^*(s) - \lambda\sum_a \pi_\lambda^*(a|s)^2\phi'(\pi_\lambda^*(a|s)),$$

*where $g_\phi(x) = (f'_\phi)^{-1}(x)$ is strictly decreasing and $\mu_\lambda^*(s)$ is a normalization term so that $\sum_{a \in \mathcal{A}} \pi_\lambda^*(a|s) = 1$.*

Theorem 1 shows how the regularization influences the optimality condition. Let $f'_\phi(0) \triangleq \lim_{x \to 0+} f'_\phi(x)$ for short. From (5), it can be shown that the optimal policy $\pi_\lambda^*$ assigns zero probability to the actions whose Q-values $Q_\lambda^*(s, a)$ are below the threshold $\mu_\lambda^*(s) - \lambda f'_\phi(0)$ and assigns positive probability to near optimal actions in proportion to their Q-values (since $g_\phi(x)$ is decreasing). The threshold involves $\mu_\lambda^*(s)$ and $f'_\phi(0)$. $\mu_\lambda^*(s)$ can be uniquely solved from the equation obtained by plugging Eqn.(5) into $\sum_{a \in \mathcal{A}} \pi_\lambda^*(a|s) = 1$. Note that the resulting equation only involves $\{Q_\lambda^*(s, a)\}_{a \in \mathcal{A}}$. Thus $\mu_\lambda^*(s)$ is actually always a multivariate finite-valued function of $\{Q_\lambda^*(s, a)\}_{a \in \mathcal{A}}$. However, the value $f'_\phi(0)$ can be infinity, making the threshold out of function. To see this, if $f'_\phi(0) = \infty$, the threshold will be

$-\infty$ and all actions will be assigned positive probability in any optimal policy. To characterize the number of zero probability actions, we define a $\delta$-sparse policy as Definition 1 shows. It is trivial that $\frac{1}{|\mathcal{A}|} \leq \delta \leq 1$. For instance, a deterministic optimal policy in non-regularized MDP is $\frac{1}{|\mathcal{A}|}$-sparse.

**Definition 1** *A given policy* $\pi : \mathcal{S} \to \Delta_\mathcal{A}$ *is called* $\delta$-sparse *if it satisfies:*

$$\frac{|\{(s,a) \in \mathcal{S} \times \mathcal{A} | \pi(a|s) \neq 0\}|}{|\mathcal{S}||\mathcal{A}|} \leq \delta. \tag{6}$$

*If* $\pi(a|s) > 0$ *for all* $(s,a) \in \mathcal{S} \times \mathcal{A}$, *we call it has no sparsity.*

**Theorem 2** *If* $\lim\limits_{x \to 0+} f'_\phi(x) = \infty$ *(or* $0 \notin \mathrm{dom} f'_\phi$*), the optimal policy of regularized MDP is not sparse.*

Theorem 2 provides us a criteria to determine whether a regularization could render its corresponding regularized optimal policy the property of sparseness. To facilitate understanding, let us see two examples. When $\phi(x) = -\log(x)$, we have that $\lim\limits_{x \to 0+} f'_\phi(x) = \lim\limits_{x \to 0+} -\log(x) - 1 = \infty$, which implies that the optimal policy of Shannon entropy-regularized MDP does not have sparsity. When $\phi(x) = \frac{k}{q-1}(1 - x^{q-1})$ for $q > 1$ and $\lambda$ is small enough, the corresponding optimal policy can be spare if $\lambda$ is small enough because $\lim\limits_{x \to 0+} f'_\phi(x) = \frac{k}{q-1}$. What's more, the sparseness property of Tsallis entropy still keeps for $1 < q < \infty$ and small $\lambda$, which is empirically proved true in [20]. Additionally, when $0 < q < 1$, the Tsallis entropy could no longer lead to sparseness due to $\lim\limits_{x \to 0+} f'_\phi(x) = \lim\limits_{x \to 0+} \frac{k}{1-q}(qx^{q-1} - 1) = \infty$.

The sparseness property is first discussed in [19] which shows the Tsallis entropy with $k = \frac{1}{2}$ and $q = 2$ can devise a sparse MDP. However, we point out that any $\phi(\cdot)$ such that $f'_\phi(0) < \infty$ with a proper $\lambda$ leads to a *sparse* MDP. Let $\mathcal{G} \subseteq \mathcal{F}$ be the set that satisfies $\forall \phi \in \mathcal{G}, 0 \in \mathrm{dom} f'_\phi$. The positive combination of any two regularizers belonging to $\mathcal{G}$ still belongs to $\mathcal{G}$.

**Proposition 2** *Given* $\phi_1, \phi_2 \in \mathcal{G}$, *we have that* $\alpha\phi_1 + \beta\phi_2 \in \mathcal{G}$ *for all* $\alpha, \beta \geq 0$. *However, if* $\phi_1 \in \mathcal{G}$ *but* $\phi_2 \notin \mathcal{G}$, $\alpha\phi_1 + \beta\phi_2 \notin \mathcal{G}$ *for any positive* $\beta$.

It is easily checked that the two families (i.e., exponential functions and trigonometric functions) given in Section 3.1 can also induce a sparse MDP with a proper $\lambda$. For convenience, we prefer to term the function $\phi(x) = \frac{k}{q-1}(1 - x^{q-1})$ that defines the Tsallis entropy as a *polynomial* function, because when $q > 1$ it is a polynomial function of degree $q-1$. Additionally, these three *basic* families of functions could be combined to construct more complex regularizers (Propositions 1, 2).

**Control the Sparsity of Optimal Policy.** Theorem 2 shows $0 \in \mathrm{dom} f'_\phi$ is necessary but not sufficient for that the optimal policy $\pi^*_\lambda$ is sparse. The sparsity of optimal policy is also controlled by $\lambda$. Theorem 3 shows how the sparsity of optimal policy can be controlled by $\lambda$ when $f'_\phi(0) < \infty$. The proof is detailed in Appendix E.

**Theorem 3** *Let* $Q^*_\lambda(s,a)$ *and* $\mu^*_\lambda(s)$ *be defined in Theorem 1 and assume* $f'_\phi(0) < \infty$. *When* $\lambda \to 0$, *the sparsity of the optimal policy* $\pi^*_\lambda$ *shrinks to* $\delta = \frac{1}{|\mathcal{A}|}$. *When* $\lambda \to \infty$, *the optimal policy has no sparsity. More specifically,* $\pi^*_\lambda(a|s) \to \frac{1}{|\mathcal{A}|}$ *for all* $(s,a) \in \mathcal{S} \times \mathcal{A}$ *as* $\lambda \to \infty$.

## 3.3 Properties of regularized MDPs

In this section, we present some properties of regularized MDPs. We first prove the uniqueness of the optimal policy and value. Next, we give the bound of the performance error between $\pi^*_\lambda$ (the optimal policy obtained by a regularized MDP) and $\pi^*$ (the policy obtained by the original MDP). In the proofs of this section, we need an additional assumption for regularizers. Assumption 2 is quite weak. All the functions introduced in Section 3.1 satisfy it.

**Assumption 2** *The regularizer* $\phi(\cdot)$ *satisfies* $f_\phi(0) \triangleq \lim\limits_{x \to 0^+} x\phi(x) = 0$.

**Generic Bellman Operator** $\mathcal{T}_\lambda$ We define a new operator $\mathcal{T}_\lambda$ for regularized MDPs, which defines a smoothed maximum. Given one state $s \in \mathcal{S}$ and current value function $V_\lambda$, $\mathcal{T}_\lambda$ is defined as

$$\mathcal{T}_\lambda V_\lambda(s) \triangleq \max_\pi \sum_a \pi(a|s) \left[Q_\lambda(s,a) + \lambda\phi(\pi(a|s))\right], \tag{7}$$

where $Q_\lambda(s,a) = r(s,a) + \gamma\mathbb{E}_{s'|s,a}V_\lambda(s')$ is Q-value function derived from one-step foreseeing according to $V_\lambda$. By definition, $\mathcal{T}_\lambda$ maps $V_\lambda(s)$ to its possible highest value which considers both future discounted rewards and regularization term. We provide simple upper and lower bounds of $\mathcal{T}_\lambda$ w.r.t. $\mathcal{T}$, i.e.,

**Theorem 4** *Under Assumptions 1 and 2, for any value function $V$ and $s \in \mathcal{S}$, we have*

$$\mathcal{T}V(s) \leq \mathcal{T}_\lambda V(s) \leq \mathcal{T}V(s) + \lambda\phi(\frac{1}{|\mathcal{A}|}). \tag{8}$$

The bound (8) shows that $\mathcal{T}_\lambda$ is a bounded and smooth approximation of $\mathcal{T}$. When $\lambda = 0$, $\mathcal{T}_\lambda$ degenerates to the Bellman operator $\mathcal{T}$. Moreover, it can be proved that $\mathcal{T}_\lambda$ is a $\gamma$-contraction. By the Banach fixed point theorem [35], $V_\lambda^*$, the fixed point of $\mathcal{T}_\lambda$, is unique. As a result of Theorem 1, $Q_\lambda^*$ and $\pi_\lambda^*$ are both unique. We formally state the conclusion and give the proof in Appendix C.

**Performance Error Between $V_\lambda^*$ and $V^*$** In general, $V^* \neq V_\lambda^*$. But their difference is controlled by both $\lambda$ and $\phi(\cdot)$. The behavior of $\phi(x)$ around the origin represents the regularization ability of $\phi(x)$. Theorem 5 shows that when $|\mathcal{A}|$ is quite large (which means $\phi(\frac{1}{|\mathcal{A}|})$ is close to $\phi(0)$ due to its continuity), the closeness of $\phi(0)$ to 0 also determines their difference. As a result, the Tsallis entropy regularized MDPs have always tighter error bounds than the Shannon entropy regularized MDPs, because the value at the origin of the concave function $\frac{k}{q-1}(1 - x^{q-1})(q > 1)$ is much lower than that of $-\log x$, both function satisfying in Assumption 2. Our theory incorporates the result of Lee et al. [19, 20] which shows a similar performance error for (general) Tsallis entropy RL. The proof of Theorem 5 is detailed in Appendix D.

**Theorem 5** *Under Assumptions 1 and 2, the error between $V_\lambda^*$ and $V^*$ can be bounded as*

$$\|V_\lambda^* - V^*\|_\infty \leq \frac{\lambda}{1-\gamma}\phi(\frac{1}{|\mathcal{A}|}).$$

## 4 Regularized Actor-Critic

To solve the problem (2) in complex environments, we propose an off-policy algorithm *Regularized Actor-Critic* (RAC), which alternates between policy evaluation and policy improvement. In practice, we apply neural networks to parameterize the Q-value and policy to increase expressive power. In particular, we model the regularized Q-value function $Q_\theta(s,a)$ and a tractable policy $\pi_\psi(a|s)$. We use Adam [17] to optimize $\psi, \theta$. Actually, RAC is created by consulting the previous work SAC [13, 14] and making some necessary changes so that it is able to be agnostic to the form of regularization.

The goal for training regularized Q-value parameters is to minimize the general Bellman residual:

$$J_Q(\theta) = \frac{1}{2}\hat{\mathbb{E}}_\mathcal{D}(Q_\theta(s_t, a_t) - y)^2, \tag{9}$$

where $\mathcal{D}$ is the replay buffer used to eliminate the correlation of sampled trajectory data and $y$ is the target function defined as follows

$$y = r(s_t, a_t) + \gamma\left[Q_{\bar\theta}(s_{t+1}, a_{t+1}) + \lambda\phi(\pi_\psi(a_{t+1}|s_{t+1}))\right].$$

The target involves a target regularized Q-value function with parameters $\bar\theta$ that are updated in a moving average fashion, which can stabilize the training process [24, 13]. Thus the gradient of $J_Q(\theta)$ w.r.t. $\theta$ can be estimated by

$$\hat\nabla J_Q(\theta) = \hat{\mathbb{E}}_\mathcal{D}\nabla_\theta Q_\theta(s_t, a_t)\left(Q_\theta(s_t, a_t) - y\right).$$

For training policy parameters, we minimize the negative total reward:

$$J_\pi(\psi) = \hat{\mathbb{E}}_\mathcal{D}\left[\mathbb{E}_{a\sim\pi_\psi(\cdot|s_t)}\left[-\lambda\phi(\pi_\psi(a|s_t)) - Q_\theta(s_t, \phi(\pi_\psi(a|s_t)))\right]\right]. \tag{10}$$

RAC is formally described in Algorithm 1. The method alternates between data collection and parameter updating. Trajectory data is collected by executing the current policy in environments and then stored in a replay buffer. Parameters of the function approximators are updated by descending along the stochastic gradients computed from the batch sampled from that replay buffer. The method makes use of two Q-functions to overcome the positive bias incurred by overestimation of Q-value, which is known to yield a poor performance [15, 9]. Specifically, these two Q-functions are parametrized by different parameters $\theta_i$ and are independently trained to minimize $J_Q(\theta_i)$. The minimum of these two Q-functions is used to compute the target value $y$ which is involved in the computation of $\hat{\nabla} J_Q(\theta)$ and $\hat{\nabla} J_\pi(\psi)$.

## 5 Experiments

We investigate the performance of different regularizers among diverse environments. We first test basic and combined regularizers in two numerical environments. Then we test basic regularizers in Atari discrete problems. In the end, we explore the possible application in Mujoco control environments.

### 5.1 Numerical results

The two discrete numerical environments we consider include a simple random generated MDP ($S = 50, A = 10$) and a Gridworld environment ($S = 81, A = 4$). Refer to Appendix H.1 for more detail settings.

**Regularizers.** Four *basic* regularizers include `shannon` ($-\log x$), `tsallis` ($\frac{1}{2}(1-x)$), `cos` ($\cos(\frac{\pi}{2}x)$) and `exp` ($\exp(1) - \exp(x)$). Proposition 1 and 2 allow three combined regularizers: (1) `min`: the minimum of `tsallis` and `shannon`, i.e., $\min\{-\log(x), 2(1-x)\}$, (2) `poly`: the positive addition of two polynomial functions, i.e., $\frac{1}{2}(1-x) + (1-x^2)$, and (3) `mix`: the positive addition of `tsallis` and `shannon`, i.e., $-\log(x) + \frac{1}{2}(1-x)$.

---

**Algorithm 1** Regularized Actor-Critic (RAC)

**Input:** $\theta_1, \theta_2, \psi$
**Initialization:** $\bar{\theta}_1 \leftarrow \theta_1, \bar{\theta}_1 \leftarrow \theta_2, \mathcal{D} \leftarrow \emptyset$
**for** each iteration **do**
  **for** each environment step **do**
    sample action, $a_t \sim \pi_\psi(\cdot|s_t)$
    receive reward $r_t \sim r_t(s_t, a_t)$
    receive next state $s_{t+1}$ from environment
    $\mathcal{D} \leftarrow \mathcal{D} \bigcup \{(s_t, a_t, r_t, s_{t+1})\}$
  **end for**
  **for** each gradient step **do**
    $\theta_i \leftarrow \theta_i - \eta_Q \hat{\nabla} J_Q(\theta_i)$ for $i \in \{1, 2\}$
    $\psi \leftarrow \psi - \eta_\pi \hat{\nabla} J_\pi(\psi)$
    $\bar{\theta}_i \leftarrow \tau\theta_i + (1-\tau)\bar{\theta}_i$ for $i \in \{1, 2\}$
  **end for**
**end for**
**Output:** $\theta_1, \theta_2, \psi$

---

**Sparsity and Convergence.** From (a)(b) in Figure 1, when $\lambda$ is extremely large, $\delta = 1$ for all regularizers. (c) shows how the probability of each action in the optimal policy at a given state varies with $\lambda$ (one curve represents one action). These results validate the Theorem 3. A reasonable explanation is that large $\lambda$ reduces the importance of discounted reward sum and makes $H_\phi(\pi)$ dominate the loss, which forces the optimal policy to put probability mass evenly on all actions in order to maximize $H_\phi(\pi)$. We regard the ability to defend the tendency towards converging to a uniform distribution as sparseness power. From our additional experiments in Appendix H, `cos` has the strongest sparseness power. (d) shows the convergence speed of RPI on different regularizers. It also shows that $\|V^* - V^{\pi^*_\lambda}\|_\infty$ is bounded as Theorem 4 states.

### 5.2 Atari results

**Regularizers.** We test four basic regularizers across four discrete control tasks from OpenAI Gym benchmark [5]. All the training details are in Appendix H.2.

**Performance.** Figure 2 shows the score during training for RAC with four regularization forms with best performance over $\lambda = \{0.01, 0.1, 1.0\}$. Except Breakout, `Shannon` performs worse than other three regularizers. `Cos` performs best in Alien and Seaquest while `tsallis` performs best in Boxing and `exp` performs quite normally. Appendix H.2 gives all the results with different $\lambda$ and sensitive analysis. In general, `shannon` is the most insensitive among others.

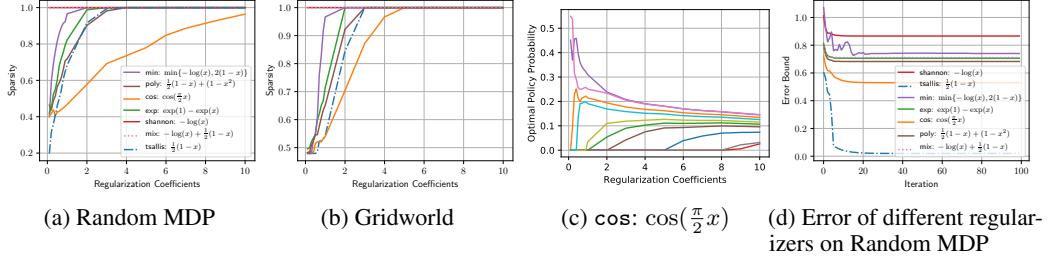

| (a) Random MDP | (b) Gridworld | (c) cos: $\cos(\frac{\pi}{2}x)$ | (d) Error of different regularizers on Random MDP |

Figure 1: (a) and (b) show the results of the sparsity $\delta$ of optimal policies on Random MDP and Gridworld. (c) shows the changing process of the probability of each action in optimal policy regularized by $\cos(\frac{\pi}{2}x)$ on Random MDP. (d) shows the $\ell_\infty$-error between $V^*$ and $V^{\pi^*_\lambda}$.

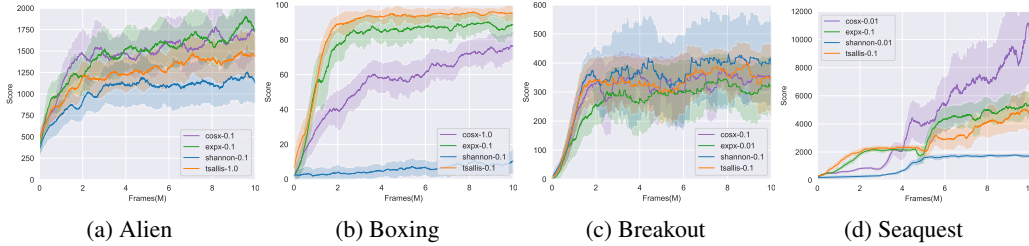

| (a) Alien | (b) Boxing | (c) Breakout | (d) Seaquest |

Figure 2: Training curves on Atari games. Each entry in the legend is named with the rule the regularization form $+\lambda$. The score is smoothed with 100 windows while the shaded area is the one standard deviation.

## 5.3 Mujoco results

**Regularizers.** We explore basic regularizers across four continuous control tasks from OpenAI Gym benchmark [5] with the MuJoCo simulator [38]. Unfortunately cos is quite unstable and prone to gradient exploding problems in deep RL training process. We speculate it instableness roots in numerical issues where the probability density function often diverges into infinity. What's more, the periodicity of $\cos(\frac{\pi}{2}x)$ makes the gradients vacillate and the algorithm hard to converge. All the details of the following experiments are given in Appendix H.3.

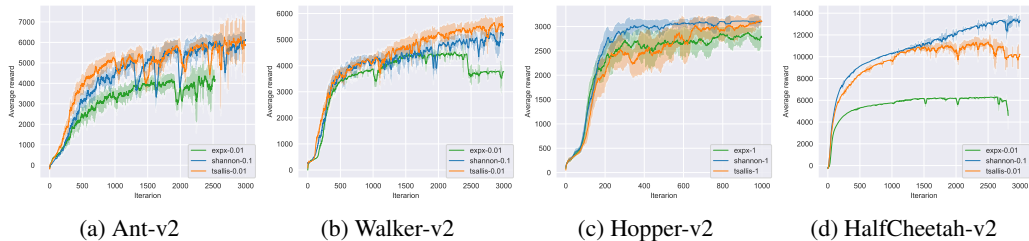

| (a) Ant-v2 | (b) Walker-v2 | (c) Hopper-v2 | (d) HalfCheetah-v2 |

Figure 3: Training curves on continuous control benchmarks. Each curve is the average of four experiments with different seeds. Each entry in the legend is named with the rule the regularization form $+\lambda$. The score is smoothed with 30 windows while the shaded area is the one standard deviation.

**Performance.** Figure 3 shows the total average return of rollouts during training for RAC with three regularization forms and different regularization coefficients ($[0.01, 0.1, 1]$). For each curve, we train four different instances with different random seeds. Tsallis performs steadily better than shannon given the same regularization coefficient $\lambda$. Tsallis is also more stable since its shaded area is thinner than shannon. Exp performs almost as good as tsallis in Ant-v2 and Hopper-v2 but performs badly in the rest two environments. From the sensitivity analysis provided in Appendix H.3, tsallis is less sensitive to $\lambda$ than cos and shannon.

# 6 Conclusion

In this paper, we have proposed a unified framework for regularized reinforcement learning, which includes entropy-regularized RL as a special case. Under this framework, the regularization function characterizes the optimal policy and value of the corresponding regularized MDPs. We have shown there are many regularization functions that can lead to a sparse but multi-modal optimal policy such as trigonometric and exponential functions. We have specified a necessary and sufficient condition for these regularization functions that could lead to sparse optimal policies and how the sparsity is controlled with $\lambda$. We have presented the logical and mathematical foundations of these properties and also conducted the experimental results.

## Acknowledgements

This work is sponsored by the Key Project of MOST of China (No. 2018AAA0101000), by Beijing Municipal Commission of Science and Technology under Grant No. 181100008918005, and by Beijing Academy of Artificial Intelligence (BAAI).

## Footnotes

[2]The general Tsallis entropy is defined with an additional real-valued parameter, called an entropic index. Lee et al. [20] shows that when this entropic index in large enough, the optimal policy is sparse.

[3]$\pi^*$ is not necessarily deterministic. If there are two actions $a_1, a_2$ that obtain the maximum of $\mathcal{T}V(s)$ for a fixed $s \in \mathcal{S}$, one can show that the stochastic policy $\pi(a_1|s) = 1 - \pi(a_2|s) = p \in [0, 1]$ is also optimal.

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
