[Supplementary Material · appendix.pdf]

# Appendix

## A   Related Work

**Regularization in RL.**   The first class aims to control the complexity of value function approximation. The use of function approximation makes it possible to model value (or Q-value) function when the state space is large or even infinite. The main regularization form is $L_2$ or $L_1$ regularization. For example, [23, 7] uses $L_2$ regularization to control the complexity of fitting value (or Q-value) functions. [18, 16] uses $L_1$ regularization for sparse feature selection.

The second class aims to capture the geometry of parameter spaces better and confine the information loss of policy search [30]. A lot of works propose to constraint the updated policy $\pi_{\text{new}}$ so that it is *close* to the old one $\pi_{\text{old}}$ in some sense. [30, 32, 22, 34, 21] use the Kullback-Leibler (KL) divergence as the measure for closeness and [3] considers a more general class of f-divergences.

The third class aims to modify the original MDP to a more tractable one. One considers the case the transition probabilities can be rescaled [37]. Others add a policy-related regularization term to the rewards, where entropy-regularized RL belongs. [29, 25, 12, 13] consider using the Shannon entropy, which is shown to improve both exploration and robustness. An MDP with Shannon entropy maximization is termed as *soft MDP* where the hard max operator is replaced by a softmax [1]. However, the optimal policy in soft MDPs put probability mass on all actions, implying some significantly unimportant actions would be executed. To fix this problem, [11] proposes to dynamically learn a prior that weights the importance of actions by using mutual information. Alternatively, [27, 19] replace Shannon entropy with Tsallis entropy, since a special case ($q = 2$ in our notation) of Tsallis entropy can devise a sparse optimal policy [19]. Recently, [20] analyzes a more general Tsallis entropy family with an additional real-valued parameter (i.e., $q$ mentioned above), called an *entropic index*, which is able to control the exploration tendency. [10] considers a more general class of regularized MDP where any strongly concave function replaces the entropy-like regularization term.

To address the issues discussed in the introduction (i.e., to obtain a sparse but multi-modal optimal policy), only the regularization in the third class could work. However, they either focus on entropy regularization or consider too large function, the former ignoring various regularization forms in convex optimization and the latter having no implications for the choice of regularization. Thus we are motivated to propose a unified framework for regularized RL which extends current entropy-regularized RL and provides enough practical guidance.

**Optimization for Entropy-regularized MDPs.**   In the literature, there are many algorithms to solve entropy-regularized MDP problems. Similarly, these methods can be modified to solve regularized MDPs since the regularization we proposed is an extension of the traditional entropy.

[12, 19] consider the general modified value iteration approach. They repeatedly solve greedily the target regularized Q-values and updates the Q-value function in a Q-learning-like pattern. [33] discussed the equivalence between policy gradients and Q-learning where the entropy regularizer is Shannon entropy. [13] adopted actor-critic methods to solve the Shannon regularized MDP in an off-policy fashion and achieves the state-of-the-art performance in continuous control tasks in RL. [20] proposes TAC, a variant of SAC, by replacing Shannon entropy with general Tsallis entropy. [25] point out there exists a path consistency equation which only the (near) optimal value and policy satisfy and propose to minimize the residual of that equation by simultaneously updating value and policy functions. This method is called as *Path Consistency Learning*(PCL). [26, 27, 6] share the same methodology with PCL for Shannon entropy.

[28] provides a unified view of entropy-regularized MDPs which enables us to formalize most entropy-regularized RL algorithms as approximate variants of Mirror Descent or Dual Averaging. [10] extends this result such that a broader class of regularizers is considered. They propose a modified policy iteration and give error propagation analyses for many existing algorithmic schemes.

# B   Proof for Optimality Condition of Regulazied MDPs

In this section, we give the detail proof for Theorem 1, which states the optimality condition of regularized MDPs. The proof follows from the Karush-Kuhn-Tucker (KKT) conditions where the derivative of a Lagrangian objective function with respect to policy $\pi(a|s)$ is set zero. Hence, our main theory is necessary and sufficient.

**Proof for Theorem 1** The Lagrangian function of (2) obtained by the optimal policy is written as follows

$$L(\pi, \beta, \mu) = \sum_s d_\pi(s) \sum_a \pi(a|s) \left( Q_\lambda^*(s,a) + \lambda \phi(\pi(a|s)) \right)$$

$$- \sum_s d_\pi(s)[\mu(s)(\sum_a \pi(a|s) - 1) + \sum_a \beta(a|s)\pi(a|s)]$$

where $d_\pi$ is the stationary state distribution of the policy $\pi$, $\mu$ and $\beta$ are Lagrangian multipliers for the equality and inequality constraints respectively. Let $f_\phi(x) = x\phi(x)$. Then the KKT condition of (2) are as follows, for all states and actions

$$0 \leq \pi(a|s) \leq 1 \text{ and } \sum_a \pi(a|s) = 1 \tag{11}$$

$$0 \leq \beta(a|s) \tag{12}$$

$$\beta(a|s)\pi(a|s) = 0 \tag{13}$$

$$Q_\lambda^*(s,a) + \lambda f_\phi'(\pi(a|s)) - \mu(s) + \beta(a|s) = 0 \tag{14}$$

where (11) is the feasibility of the primal problem, (12) is the feasibility of the dual problem, (13) results from the complementary slackness and (14) is the stationarity condition. We eliminate $d_\pi(s)$ since we assume all policies induce an irreducible Markov chain.

Since $f_\phi(x) = x\phi(x)$ is a strictly decreasing function due to (4) in Assumption 1, its inverse function $g_\phi(x) = (f_\phi')^{-1}(x)$ is also strictly decreasing. From (14), we can resolve $\pi(a|s)$ as

$$\pi(a|s) = g_\phi \left( \frac{1}{\lambda}(\mu(s) - Q_\lambda^*(s,a) - \beta(a|s)) \right). \tag{15}$$

Fix a state $s$. For any positive action, its corresponding Lagrangian multiplier $\beta(a|s)$ is zero due to the complementary slackness and $Q_\lambda^*(s,a) > \mu(s) - \lambda f_\phi'(0)$ must hold. For any zero-probability action, its Lagrangian multiplier $\beta(a|s)$ will be set such that $\pi(a|s) = 0$. Note that $\beta(a|s) \geq 0$, thus $Q_\lambda^*(s,a) \leq \mu(s) - \lambda f_\phi'(0)$ must hold in this case. From these observations, $\pi(a|s)$ can be reformulated as

$$\pi(a|s) = \max \left\{ g_\phi \left( \frac{1}{\lambda}(\mu(s) - Q_\lambda^*(s,a)) \right), 0 \right\} \tag{16}$$

By plugging (16) into (11), we obtain an new equation

$$\sum_a \max \left\{ g_\phi \left( \frac{1}{\lambda}(\mu(s) - Q_\lambda^*(s,a)) \right), 0 \right\} = 1 \tag{17}$$

Lemma 1 states that (17) has and only has one solution denoted as $\mu_\lambda^*$. Therefore, $\mu_\lambda^*$ can be solved uniquely. We defer the proof of Lemma 1 later in this section.

Next we aim to obtain the optimal state value $V_\lambda^*$. It follows that

$$V_\lambda^*(s) = \mathcal{T}_\lambda V_\lambda^*(s)$$

$$= \sum_a \pi_\lambda^*(a|s) \left( Q_\lambda^*(s,a) + \lambda \phi(\pi_\lambda^*(a|s)) \right)$$

$$= \sum_a \pi_\lambda^*(a|s) \left( \mu_\lambda^*(s) - \lambda \pi_\lambda^*(a|s)\phi(\pi_\lambda^*(a|s)) \right)$$

$$= \mu_\lambda^*(s) - \lambda \sum_a \pi_\lambda^*(a|s)^2 \phi'(\pi_\lambda^*(a|s)).$$

The first equality follows from the definition of the optimal state value. The second equality holds because $\pi_\lambda^*$ maximizes $\mathcal{T}_\lambda V_\lambda^*(s)$. The third equality results from plugging (14).

To summarize, we obtain the optimality condition of regularized MDPs as follows

$$Q_\lambda^*(s,a) = r(s,a) + \gamma \mathbb{E}_{s'|s,a} V_\lambda^*(s'),$$

$$\pi_\lambda^*(a|s) = \max\left\{g_\phi\left(\frac{1}{\lambda}(\mu_\lambda^*(s) - Q_\lambda^*(s,a))\right), 0\right\},$$

$$V_\lambda^*(s) = \mu_\lambda^*(s) - \lambda \sum_a \pi_\lambda^*(a|s)^2 \phi'(\pi_\lambda^*(a|s)),$$

where $g_\phi(x) = (f_\phi')^{-1}(x)$ is strictly decreasing and $\mu_\lambda^*(s)$ is a normalization term so that $\sum_{a \in \mathcal{A}} \pi_\lambda^*(a|s) = 1$. ∎

**Lemma 1** *For any Q-value function $Q(s,a)$, the equation*

$$\sum_a \max\left\{g_\phi\left(\frac{1}{\lambda}(\mu(s) - Q(s,a))\right), 0\right\} = 1 \tag{18}$$

*has and only has one $\mu^*$ satisfying it.*

**Proof** Denote the left hand side of (18) which is a continuous function of $\mu$ as $h(\mu)$. We first prove that $h(\mu)$ is a strictly decreasing function on $(-\infty, \mu_{\max})$, where $\mu_{\max} = \max_a Q(s,a) + \lambda f_\phi'(0)$. Let $\Lambda(s,\mu)$ the set of actions such that their maximum term in (18) is not obtained at 0, i.e., $\Lambda(s,\mu) = \{a : Q(s,a) > \mu(s) - \lambda f_\phi'(0)\}$. Then for $\mu_1 < \mu_2 < \mu_{\max}$, it follows that $\Lambda(s,\mu_2) \subseteq \Lambda(s,\mu_1)$ and

$$h(\mu_1) - h(\mu_2) = \sum_{a \in \Lambda(s,\mu_2)} \Delta(\mu_1, \mu_2) + \sum_{a \in \Lambda(s,\mu_1) - \Lambda(s,\mu_2)} g_\phi\left(\frac{1}{\lambda}(\mu_1(s) - Q(s,a))\right)$$

where

$$\Delta(\mu_1, \mu_2) = g_\phi\left(\frac{1}{\lambda}(\mu_1(s) - Q(s,a))\right) - g_\phi\left(\frac{1}{\lambda}(\mu_2(s) - Q(s,a))\right)$$

is positive for all actions in $\Lambda(s,\mu_2)$. Since there must be at least one action in $\Lambda(s,\mu_2)$, $h(\mu_1) - h(\mu_2) > 0$. Therefore, we have proved that $h(\mu)$ decreases strictly on $(-\infty, \mu_{\max})$. Note that $h(\mu_{\max}) = 0 < 1$ and $h(\mu_{\min}) > 1$ where $\mu_{\min} = \min_a Q(s,a) + \lambda f_\phi'(1)$. This result implies there exist a unique $\mu^* \in (\mu_{\min}, \mu_{\max})$ satisfying (18) as the result of the intermediate value theorem. ∎

## C  Proof for General Bellman Operator

In (7), we define a general Bellman operator $\mathcal{T}_\lambda$ for regularized MDPs. Given one state $s \in \mathcal{S}$ and current value function $V_\lambda$,

$$(\mathcal{T}_\lambda V_\lambda)(s) := \max_\pi \sum_a \pi(a|s) \left[Q_\lambda(s,a) + \lambda\phi(\pi(a|s))\right],$$

where $Q_\lambda(s,a) = r(s,a) + \gamma \mathbb{E}_{s'|s,a} V_\lambda(s')$ is Q-value function deriving from one-step foreseeing according to $V_\lambda$. In Lemma 2, we prove $\mathcal{T}_\lambda$ is a $\gamma$-contraction. In Theorem 4, we prove the simple lower and upper bound for $\mathcal{T}_\lambda$ under Assumption 2.

**Lemma 2** $\mathcal{T}_\lambda$ *is a $\gamma$-contraction.*

**Proof** For any two state value functions $V_1$ and $V_2$, let $\pi_i$ be the policy that maximize $\mathcal{T}_\lambda V_i$, $i \in \{1, 2\}$. Then it follows that for any state $s$ in $\mathcal{S}$,

$$(\mathcal{T}_\lambda V_1)(s) - (\mathcal{T}_\lambda V_2)(s)$$

$$= \sum_a \pi_1(a|s) \left[ r(s, a) + \gamma \mathbb{E}_{s'|s,a} V_1(s') + \lambda\phi(\pi_1(a|s)) \right] - \max_\pi \sum_a \pi(a|s) \left[ r(s, a) + \gamma \mathbb{E}_{s'|s,a} V_2(s') + \lambda\phi(\pi(a|s)) \right]$$

$$\leq \sum_a \pi_1(a|s) \left[ r(s, a) + \gamma \mathbb{E}_{s'|s,a} V_1(s') + \lambda\phi(\pi_1(a|s)) \right] - \sum_a \pi_1(a|s) \left[ r(s, a) + \gamma \mathbb{E}_{s'|s,a} V_2(s') + \lambda\phi(\pi_1(a|s)) \right]$$

$$= \gamma \sum_a \pi_1(a|s) \mathbb{E}_{s'|s,a}(V_1(s') - V_2(s')) \leq \gamma\|V_1 - V_2\|_\infty.$$

By symmetry, it follows that for any state $s$ in $\mathcal{S}$,

$$(\mathcal{T}_\lambda V_2)(s) - (\mathcal{T}_\lambda V_1)(s) \leq \gamma\|V_1 - V_2\|_\infty$$

Therefore, it follows that

$$\|\mathcal{T}_\lambda V_2 - \mathcal{T}_\lambda V_1\|_\infty \leq \gamma\|V_1 - V_2\|_\infty$$

∎

**Proof for Theorem 4** Fix any value function $V$ and $s \in \mathcal{S}$. Note that $\phi(\pi(a|s))$ is non-negative due to (1) and (2) in Assumption 1. Therefore, by definition the left inequality follows from

$$\mathcal{T}_\lambda V(s) = \max_\pi \sum_a \pi(a|s) \left[ r(s, a) + \gamma \mathbb{E}_{s'|s,a} V(s') + \lambda\phi(\pi(a|s)) \right]$$

$$\geq \max_\pi \sum_a \pi(a|s) \left[ r(s, a) + \gamma \mathbb{E}_{s'|s,a} V(s') \right] = \mathcal{T}V(s).$$

For the right inequality, note that

$$\mathcal{T}_\lambda V(s) = \max_\pi \sum_a \pi(a|s) \left[ r(s, a) + \gamma \mathbb{E}_{s'|s,a} V(s') + \lambda\phi(\pi(a|s)) \right]$$

$$\leq \max_\pi \sum_a \pi(a|s) \left[ r(s, a) + \gamma \mathbb{E}_{s'|s,a} V(s') \right] + \lambda \max_\pi H_\phi(\pi)$$

$$= \mathcal{T}V(s) + \lambda \max_\pi H_\phi(\pi).$$

where $H_\phi(\pi) = \sum_a \pi(a|s)\phi(\pi(a|s))$ defined in (3) is what we next aim to bound.

The Lagrangian of solving $\max_\pi H_\phi(\pi)$ is

$$L(\pi, \beta, \mu) = H_\phi(\pi) + \mu\left(\sum_a \pi(a|s) - 1\right) + \beta_a \pi(a|s).$$

Its stationary condition is

$$\frac{\partial L}{\partial \pi(a|s)} = f'_\phi(\pi(a|s)) + \mu + \beta_a = 0.$$

If $\pi(a|s) > 0$ then $\beta_a = 0$ from the complementary slackness. Let $\pi^*$ be the policy that maximizes $H_\phi(\pi)$ and $S = \{a : \pi^*(a|s) > 0\}$ be its support set. Then $\pi(a|s) = g_\phi(-\mu) =$ constant for all $a \in S$. Hence, $\pi(a|s) = \frac{1}{|S|}$ for $a \in S$ and $= 0$ for $a \notin S$. Note that $g_\phi$ is strictly decreasing and the assumption $\lim_{x \to 0^+} x\phi(x) = 0$,

$$H_\phi(\pi) = \sum_{a \in S(s)} \pi^*(a|s)\phi(\pi^*(a|s)) = \phi\left(\frac{1}{|S|}\right) \leq \phi\left(\frac{1}{|\mathcal{A}|}\right)$$

where the last inequality use the fact $\phi$ is decreasing and $|S| \leq |\mathcal{A}|$.

∎

# D Proof for Performance Error

We prove Theorem 5 in that the difference of $V^*$ and $V^*_\lambda$ is controlled by both $\lambda$ and $\phi(\cdot)$ under Assumption 2. To that end, we first introduce several useful lemmas which give some properties of $\mathcal{T}_\lambda$ including monotonicity, translation and convergence of repeated applications. Then a combination of these lemmas will prove Theorem 5.

**Lemma 3 (Monotonicity)** $\mathcal{T}_\lambda$ *has the property of monotonicity, i.e., if* $V_1(s) \leq V_2(s)$ *for all* $s \in \mathcal{S}$, *then* $\mathcal{T}_\lambda V_1(s) \leq \mathcal{T}_\lambda V_2(s)$ *for all* $s \in \mathcal{S}$.

**Proof** The conclusion directly follows from

$$\mathcal{T}_\lambda V_1(s) = \max_\pi \sum_a \pi(a|s) \left[ r(s,a) + \gamma \mathbb{E}_{s'|s,a} V_1(s') + \lambda\phi(\pi(a|s)) \right]$$

$$\leq \max_\pi \sum_a \pi(a|s) \left[ r(s,a) + \gamma \mathbb{E}_{s'|s,a} V_2(s') + \lambda\phi(\pi(a|s)) \right] = \mathcal{T}_\lambda V_2(s)$$

∎

**Lemma 4 (Translation)** *Let* $c$ *denote any constant. Define* $(V + c)(s) \triangleq V(s) + c$ *as the value function shifted by* $c$. *Then it follows that for any* $s \in \mathcal{S}$,

$$(\mathcal{T}_\lambda(V + c))(s) = (\mathcal{T}_\lambda V)(s) + \gamma c$$

**Proof** By definition, it directly follows from

$$(\mathcal{T}_\lambda(V + c))(s) = \max_\pi \sum_a \pi(a|s) \left[ r + \gamma \mathbb{E}_{s'|s,a}(V + c)(s') + \lambda\phi(\pi(a|s)) \right]$$

$$= \max_\pi \sum_a \pi(a|s) \left[ r + \gamma \mathbb{E}_{s'|s,a} V(s') + \gamma c + \lambda\phi(\pi(a|s)) \right] = (\mathcal{T}_\lambda V)(s) + \gamma c$$

∎

**Lemma 5 (Convergence of Repeated Applications)** *For any initial value function* $V_0$, *define* $V_n = \mathcal{T}^n_\lambda V_0 \triangleq \underbrace{\mathcal{T}_\lambda \cdots \mathcal{T}_\lambda}_{n} V_0$ *as the value function resulting from* $n$ *times application of* $\mathcal{T}_\lambda$ *to* $V_0$. *Then*

$$\lim_{n\to\infty} \|V_n - V^*_\lambda\|_\infty = 0.$$

**Proof** Note that $V^*_\lambda = \mathcal{T}_\lambda V^*_\lambda$. It follows that

$$\|V_n - V^*_\lambda\|_\infty = \|\mathcal{T}_\lambda V_{n-1} - \mathcal{T}_\lambda V^*_\lambda\|_\infty \leq \gamma\|V_{n-1} - V^*_\lambda\|_\infty \leq \cdots \leq \gamma^n\|V_0 - V^*_\lambda\|_\infty.$$

The first equality follows from definition. The first inequality results from Lemma 2. The last inequality is due to $n$-times applications of the first inequality. ∎

**Proof for Theorem 5** Fix any initial value function $V_0$. We aim to use mathematical induction to prove the statement that for any $n \geq 1$, it follows for any $s \in \mathcal{S}$

$$\mathcal{T}^n V_0(s) \leq \mathcal{T}^n_\lambda V_0(s) \leq \mathcal{T}^n V_0(s) + \lambda\phi(\frac{1}{|\mathcal{A}|}) \sum_{t=0}^{n-1} \gamma^t. \tag{19}$$

When $n = 1$, (19) results from Theorem 4.

Suppose the statement holds when $n = k(k \geq 1)$. Consider the case where $n = k + 1$. First it follows that

$$\mathcal{T}^{k+1} V_0(s) \leq \mathcal{T}\mathcal{T}^k_\lambda V_0(s) \leq \mathcal{T}^{k+1}_\lambda V_0(s).$$

The first inequality follows from the hypothesis and the monotonicity of $\mathcal{T}$ (which is a special case of $\mathcal{T}_\lambda$ when $\lambda = 0$) from Lemma 3. The second inequality results from letting $V = \mathcal{T}_\lambda^k V_0$ in Theorem 4.

Second, it follows that

$$
\begin{aligned}
\mathcal{T}_\lambda^{k+1} V_0(s) &= \mathcal{T}_\lambda \mathcal{T}_\lambda^k V_0(s) \\
&\leq \mathcal{T}_\lambda(\mathcal{T}^k V_0(s) + \lambda \phi(\frac{1}{|\mathcal{A}|}) \sum_{t=0}^{k-1} \gamma^t) \\
&= \mathcal{T}_\lambda \mathcal{T}^k V_0(s) + \lambda \phi(\frac{1}{|\mathcal{A}|}) \sum_{t=1}^{k} \gamma^t \\
&\leq \mathcal{T}^{k+1} V_0(s) + \lambda \phi(\frac{1}{|\mathcal{A}|}) \sum_{t=0}^{k} \gamma^t,
\end{aligned}
$$

where the first inequality follows from the induction where $n = k$ and the monotonicity of $\mathcal{T}_\lambda$ from Lemma 3, the second equality holds by setting $V = \mathcal{T}^k V_0$ and $c = \lambda \phi(\frac{1}{|\mathcal{A}|}) \sum_{t=0}^{k-1} \gamma^t$ in Lemma 4. The last inequality results from letting $V = \mathcal{T}_\lambda^k V_0$ in Theorem 4.

Putting above results together, we prove that (19) holds when $n = k + 1$. Therefore by mathematical induction, (19) holds for any positive integer $n$. From Lemma 5, we have $V^*(s) = \lim_{n \to \infty} \mathcal{T}^n V_0(s)$ and $V_\lambda^*(s) = \lim_{n \to \infty} \mathcal{T}_\lambda^n V_0(s)$. Now let $n$ go infinity in both sides of (19), we obtain

$$
V^*(s) \leq V_\lambda^*(s) \leq V^*(s) + \frac{\lambda}{1-\gamma} \phi(\frac{1}{|\mathcal{A}|}),
$$

which proves the theorem. ∎

# E   Proof for Control the sparsity of Optimal Policy

In this section, we show that the number of positive actions can be controlled by regularization coefficient $\lambda$. Similar results about Tsallis entropy regularized MDPs can be found in [19]. However their proof focuses on a specific regularization. The proof we provide is suitable for any regularizors satisfying Assumption 1.

**Proof for Theorem 3**. At first we prove that the optimal policy will approximate uniform distribution on action space. Under such situation, it is obvious that the optimal policy will have no sparsity as $\lambda \to \infty$. Denote $H = \max_\pi H(\pi)$. For an arbitrary $\delta > 0$, there exists $\lambda_0$, such that $\forall \lambda > \lambda_0$, $|\frac{r(s,a)}{\lambda}| \leq \delta$. Next we estimate the error between $\frac{Q_\lambda^*(s,a)}{\lambda}$ and $\max_\pi \mathbb{E}[\sum_{t=1}^{\infty} \gamma^t \phi(\pi(a_t|s_t))|s_0 = s, a_0 = a] = \frac{\gamma}{1-\gamma} H$:

$$
\begin{aligned}
\frac{Q_\lambda^*(s,a)}{\lambda} - \frac{\gamma}{1-\gamma} H &= \frac{Q_\lambda^*(s,a)}{\lambda} - \max_\pi \mathbb{E}[\sum_{t=1}^{\infty} \gamma^t \phi(\pi(a_t|s_t))|s_0 = s, a_0 = a, \pi] \\
&\leq \frac{Q_\lambda^{\pi_\lambda^*}(s,a)}{\lambda} - \mathbb{E}[\sum_{t=1}^{\infty} \gamma^t \phi(\pi_\lambda^*(a_t|s_t))|s_0 = s, a_0 = a, \pi_\lambda^*] \\
&= \mathbb{E}[\sum_{t=0}^{\infty} \gamma^t \frac{r(s_t, a_t)}{\lambda}|s_0 = s, a_0 = a, \pi_\lambda^*] \\
&\leq \frac{\delta}{1-\gamma}
\end{aligned}
\tag{20}
$$

On the other hand, denote $\pi_H^* = \operatorname{argmax}_\pi H(\pi)$, we have:

$$\frac{Q_\lambda^*(s,a)}{\lambda} - \frac{\gamma}{1-\gamma}H = \frac{Q_\lambda^*(s,a)}{\lambda} - \max_\pi \mathbb{E}[\sum_{t=1}^\infty \gamma^t \phi(\pi(a_t|s_t))|s_0 = s, a_0 = a, \pi]$$

$$\geq \frac{Q_\lambda^{\pi_H^*}(s,a)}{\lambda} - \mathbb{E}[\sum_{t=1}^\infty \gamma^t \phi(\pi_H^*(a_t|s_t))|s_0 = s, a_0 = a, \pi_H^*]$$

$$= \mathbb{E}[\sum_{t=0}^\infty \gamma^t \frac{r(s_t,a_t)}{\lambda}|s_0 = s, a_0 = a, \pi_H^*]$$

$$\geq -\frac{\delta}{1-\gamma} \tag{21}$$

So $|\frac{Q_\lambda^*(s,a)}{\lambda} - \frac{\gamma}{1-\gamma}H| \leq \frac{\delta}{1-\gamma}$.

Fix any $s \in \mathcal{S}$, denote $\mu_\lambda(s)$ is the solution satisfies Equation 18:

$$1 = \sum_a \max\left\{ g_\phi\left(\frac{1}{\lambda}(\mu_\lambda(s) - Q_\lambda^*(s,a))\right), 0 \right\}$$

$$> |\mathcal{A}| \max\left\{ g_\phi\left(\frac{1}{\lambda}(\mu_\lambda(s) - \min_a Q_\lambda^*(s,a))\right), 0 \right\}$$

$$> |\mathcal{A}| g_\phi\left(\frac{1}{\lambda}(\mu_\lambda(s) - \min_a Q_\lambda^*(s,a))\right)$$

As $g_\phi$ is strictly decreasing, we have $\frac{1}{\lambda}(\mu_\lambda(s) - \min_a Q_\lambda^*(s,a)) > f_\phi'(\frac{1}{|\mathcal{A}|})$. By the same method, we can obtain that $\frac{1}{\lambda}(\mu_\lambda(s) - \max_a Q_\lambda^*(s,a)) < f_\phi'(\frac{1}{|\mathcal{A}|})$. Combining with $|\frac{Q_\lambda^*(s,a)}{\lambda} - \frac{\gamma}{1-\gamma}H| \leq \frac{\delta}{1-\gamma}$:

$$\frac{\mu_\lambda(s)}{\lambda} - \frac{\gamma}{1-\gamma}H = \frac{\mu_\lambda(s)}{\lambda} - \frac{\min_a Q_\lambda^*(s,a)}{\lambda} + \frac{\min_a Q_\lambda^*(s,a)}{\lambda} - \frac{\gamma}{1-\gamma}H$$

$$> f_\phi'(\frac{1}{|\mathcal{A}|}) - \frac{\delta}{1-\gamma}$$

$$\frac{\mu_\lambda(s)}{\lambda} - \frac{\gamma}{1-\gamma}H = \frac{\mu_\lambda(s)}{\lambda} - \frac{\max_a Q_\lambda^*(s,a)}{\lambda} + \frac{\max_a Q_\lambda^*(s,a)}{\lambda} - \frac{\gamma}{1-\gamma}H$$

$$< f_\phi'(\frac{1}{|\mathcal{A}|}) + \frac{\delta}{1-\gamma}$$

For arbitrary $s, a$, the following inequality holds:

$$\frac{\mu_\lambda(s) - Q_\lambda^*(s,a)}{\lambda} = \frac{\mu_\lambda(s)}{\lambda} - \frac{\gamma}{1-\gamma}H + \frac{\gamma}{1-\gamma}H - \frac{Q_\lambda^*(s,a)}{\lambda}$$

$$> f_\phi'(\frac{1}{|\mathcal{A}|}) - \frac{2\delta}{1-\gamma}$$

$$\frac{\mu_\lambda(s) - Q_\lambda^*(s,a)}{\lambda} = \frac{\mu_\lambda(s)}{\lambda} - \frac{\gamma}{1-\gamma}H + \frac{\gamma}{1-\gamma}H - \frac{Q_\lambda^*(s,a)}{\lambda}$$

$$< f_\phi'(\frac{1}{|\mathcal{A}|}) + \frac{2\delta}{1-\gamma}$$

$$\tag{22}$$

which concludes that $|\frac{\mu_\lambda(s)-Q_\lambda^*(s,a)}{\lambda} - f_\phi'(\frac{1}{|\mathcal{A}|})| < \frac{\delta}{1-\gamma}$. By continuity of $g_\phi$, $\forall \varepsilon > 0$, choose a proper $\delta$, $|g_\phi(\frac{\mu_\lambda(s)-Q_\lambda^*(s,a)}{\lambda}) - \frac{1}{|\mathcal{A}|}| < \varepsilon$.

Next we prove that the sparsity of optimal policy $\pi_\lambda^*$ varies as $\delta \to \frac{1}{|\mathcal{A}|}$ when $\lambda \to 0$.

For arbitrary $(s,a) \in \mathcal{S} \times \mathcal{A}$, $\varepsilon > 0$ and $\lambda > 0$, the following inequality holds:

$$0 \leq Q_\lambda^*(s,a) - Q^*(s,a) \leq Q_\lambda^{\pi_\lambda^*}(s,a) - Q^{\pi_\lambda^*}(s,a)$$
$$= \lambda \mathbb{E}[\sum_{t=1}^\infty \gamma^t \phi(\pi_\lambda^*(a_t|s_t))|s_0 = s, a_0 = a, \pi_\lambda^*]$$
$$\leq \lambda \frac{\gamma}{1-\gamma} H \tag{23}$$

Denote $G(s) = \min_{a_1,a_2 \in \mathcal{A}} |Q^*(s,a_1) - Q^*(s,a_2)|$, if $\lambda < \frac{1-\gamma}{H\gamma} G(s)$, the order of Q-values $Q^*(s,\cdot)$ is exactly the same with the order of $Q_\lambda^*(s,\cdot)$. In other words, denote $Q^*(s,a_1) < Q^*(s,a_2) < ... < Q^*(s,a_{|\mathcal{A}|})$, then $Q_\lambda^*(s,a_1) < Q_\lambda^*(s,a_2) < ... < Q_\lambda^*(s,a_{|\mathcal{A}|})$ still holds for $\lambda < \frac{1-\gamma}{H\gamma} G(s)$.

Next we prove the desired result by contradiction. For any given $a_k \in \mathcal{A}$ and $s \in \mathcal{S}$, and $\lambda < \frac{1-\gamma}{H\gamma} G(s)$, $\exists \lambda_0 < \lambda$, such that $\pi_{\lambda_0}(a_k|s) = g_\phi(\frac{\mu_{\lambda_0}(s) - Q_{\lambda_0}^*(s,a_k)}{\lambda_0}) > 0$. With the assumption, we can construct a sequence $\frac{1-\gamma}{H\gamma} G(s) > \lambda_1 > \lambda_2 > ... > \lambda_n > ...$, which satisifies $\lim_{n\to\infty} \lambda_n = 0$ and $g_\phi(\frac{\mu_{\lambda_n}(s) - Q_{\lambda_n}^*(s,a_k)}{\lambda_n}) > 0$, which is equivalent with $\mu_{\lambda_n}(s) - Q_{\lambda_n}^*(s,a_k) < \lambda_n f_\phi'(0)$ as $g_\phi$ is a strictly decreasing function. Combining with KKT conditions (14): $\mu_{\lambda_n}(s) = Q_{\lambda_n}^*(s,a_{|\mathcal{A}|}) + \lambda_n f_\phi'(\pi_{\lambda_n}^*(a_{|\mathcal{A}|}|s))$, the following inequality holds:

$$Q_{\lambda_n}^*(s,a_{|\mathcal{A}|}) - Q_{\lambda_n}^*(s,a_k) < \lambda_n(f_\phi'(0) - f_\phi'(\pi_{\lambda_n}^*(a_{|\mathcal{A}|}|s))) < \lambda_n(f_\phi'(0) - f_\phi'(1)) \tag{24}$$

By (23) and (24),

$$Q^*(s,a_{|\mathcal{A}|}) - Q^*(s,a_k) \leq Q_{\lambda_n}^*(s,a_{|\mathcal{A}|}) - Q^*(s,a_k)$$
$$\leq Q_{\lambda_n}^*(s,a_{|\mathcal{A}|}) - Q_{\lambda_n}^*(s,a_k) + \lambda_n \frac{\gamma}{1-\gamma} H$$
$$< \lambda_n(f_\phi'(0) - f_\phi'(1) + \frac{\gamma}{1-\gamma} H) \tag{25}$$

As $\lim_{n\to\infty} \lambda_n = 0$ and $f_\phi'(0) - f_\phi'(1) + \frac{\gamma}{1-\gamma} H$ is a positive constant, then the limit of right hand side of (25) is 0, which causes conflicts with that the left hand side of (25) is a positive constant. Therefore, we claim that for any given $a \in \mathcal{A}$, $s \in \mathcal{S}$ and $a \neq \arg\max Q^*(s,\cdot)$, $\exists \lambda_{a,s} > 0$, such that $\forall \lambda \leq \lambda_{a,s}$, $\pi_\lambda^*(a|s) = 0$. So for all $\lambda < \min_{a,s} \lambda_{a,s}$, the sparsity of the optimal policy $\pi_\lambda^*$ is $\delta = \frac{1}{|\mathcal{A}|}$. ∎

# F  Regularized Policy Iteration (RPI)

To solve problem (2), we introduce *Regularized Policy Iteration* (RPI), an algorithm that alternates between policy evaluation and policy improvement in the maximum regularized MDP framework. We first derive RPI on a tabular setting and show it provably converges to an optimal policy. Then we approximate RPI into a more practical algorithm which is an actor-critic method and thus named as regularized actor-critic (RAC).

The derivation of RPI stems from generalized policy iteration [36] that alternates between policy evaluation and policy improvement. In the policy evaluation step, we wish to compute the Q-value $Q_\lambda^\pi$ of a given policy $\pi$. When $\pi$ is fixed, $Q_\lambda^\pi$ can be computed iteratively by initializing any Q-value function and repeatedly applying the modified Bellman backup operator $\mathcal{T}_\lambda^\pi$ defined by

$$\mathcal{T}_\lambda^\pi Q_\lambda(s,a) \triangleq r(s,a) + \gamma \mathbb{E}_{s'|s,a} V_\lambda(s'), \tag{26}$$

where $V_\lambda$ is the state value function derived from $Q_\lambda$,

$$V_\lambda(s') = \mathbb{E}_{a' \sim \pi(\cdot|s')}[Q_\lambda(s',a') + \phi(\pi(a'|s'))]. \tag{27}$$

One can show that by repeatedly applying $\mathcal{T}_\lambda^\pi$ to any initialized value function, the regualrized Q-value $Q_\lambda^\pi$ of the policy $\pi$ will be obtained.

In the policy improvement step, we wish to update the evaluated policy $\pi_{\text{old}}$ to an improved policy $\pi_{\text{new}}$ in terms of its regularized Q-values. Therefore for each state $s$ we update the policy according to

$$\pi_{\text{new}}(a|s) = \arg\max_{\pi} \mathbb{E}_{a\sim\pi(\cdot|s)}[Q_\lambda^{\pi_{\text{old}}}(s,a) + \lambda\phi(\pi(a|s))]. \tag{28}$$

If $\phi$ is good enough, we can find a closed form of $\pi_{\text{new}}$ for problem (28). For example, for Shannon entropy [25] with $\phi(x) = -\log(x)$, $\pi_{\text{new}}(a|s) \propto \exp(\frac{Q_\lambda^{\pi_{\text{old}}}}{\lambda})$; for Tsallis entropy [19] with $\phi(x) = \frac{1}{2}(1-x)$, $\pi_{\text{new}}(a|s) = \max\left(\frac{Q_\lambda^{\pi_{\text{old}}}(s,a)}{\lambda} - \tau(\frac{Q_\lambda^{\pi_{\text{old}}}(s,\cdot)}{\lambda}), 0\right)$, where $\tau(\frac{Q_\lambda^{\pi_{\text{old}}}(s,\cdot)}{\lambda}) = \frac{\sum_{a\in S(s,\lambda)} \frac{Q_\lambda^{\pi_{\text{old}}}(s,a)}{\lambda} - 1}{|S(s,\lambda)|}$ is the normalization term and $S(s,\lambda)$ is the number of non-zero probability state-action pair. However, for a general $\phi$, it is unlikely to find a closed form of $\pi_{\text{new}}$. In that case the solution can be obtained through a numerical optimization method, since the maximization problem (28) is a convex optimization whose domain is the probability simplex $\Delta_{\mathcal{A}}$ and traditional convex solvers could solve it efficiently. Actually, in the experiments of RPI, for the regularization forms introduced in Section 3.1 except the two examples mentioned above, there is no closed form and we use numerical optimization to improve the old policy.

Once evaluating and improving the current policy $\pi_{\text{old}}$, we can prove the resulting policy $\pi_{\text{new}}$ has a higher regularized Q-value than that of the old one. Therefore, by alternating the policy evaluation and the policy improvement, any initializing policy will provably converge to the optima policy $\pi_\lambda^*$ under the framework of regularized MDPs (Theorem 6).

**Theorem 6** *For any policy $\pi_0$, by repeatedly applying policy evaluation and regularized policy improvement, $\pi_0$ will converge to the optimal policy $\pi_\lambda^*$ in the sense that $Q_\lambda^{\pi_\lambda^*}(s,a) \geq Q_\lambda^\pi(s,a)$ for all $\pi$ and $s \in \mathcal{S}, a \in \mathcal{A}$.*

## G  Proof for Regularized Policy Iteration

In this section, we give the proof of convergence of RPI. We first that repeatedly applying $\mathcal{T}_\lambda^\pi$ to any initialized policy leads to the Q-value of a given policy $\pi_{\text{old}}$. Then we prove the policy improvement step will lead to a new policy $\pi_{\text{new}}$ which has higher Q-value than $\pi_{\text{old}}$.

**Lemma 6 (Policy Evaluation)** *Fix any policy $\pi$. Consider the Bellman backup operator $\mathcal{T}_\lambda^\pi$ in (26), for any initial Q-value $Q_0$, let $Q_n = \mathcal{T}_\lambda^\pi Q_{n-1}(n \geq 1)$. Then $\lim_{n\to\infty} \|Q_n - Q_\lambda^\pi\|_\infty = 0$.*

**Proof** Similar to Lemma 2, we can prove $\mathcal{T}_\lambda^\pi$ is a $\gamma$ contraction. Note that $Q_\lambda^\pi = \mathcal{T}_\lambda^\pi Q_\lambda^\pi$. Therefore we have that

$$\|Q_n - Q_\lambda^\pi\|_\infty = \|\mathcal{T}_\lambda^\pi Q_{n-1} - \mathcal{T}_\lambda^\pi Q_\lambda^\pi\|_\infty \leq \gamma\|Q_{n-1} - Q_\lambda^\pi\|_\infty \leq \cdots \leq \gamma^n\|Q_0 - Q_\lambda^\pi\|_\infty.$$

When $n$ goes infinity, $Q_n$ will converge to the regularized Q-value of $\pi$. ∎

**Lemma 7 (Policy Improvement)** *Let $\pi_{old}$ be the evaluated policy with $Q_\lambda^{\pi_{old}}$ its regularized Q-value and $\pi_{new}$ be the optimizer of the maximization problem defined in (28). Then $Q_\lambda^{\pi_{old}}(s,a) \leq Q_\lambda^{\pi_{new}}(s,a)$ for all $s \in \mathcal{S}$ and $a \in \mathcal{A}$.*

**Proof** Since $\pi_{\text{new}}$ is the maximizer of the problem defined in (28), it follows that for all states and actions

$$\mathbb{E}_{a\sim\pi_{\text{old}}}[Q_\lambda^{\pi_{\text{old}}}(s,a) + \lambda\phi(\pi_{\text{old}}(a|s))] \leq \mathbb{E}_{a\sim\pi_{\text{new}}}[Q_\lambda^{\pi_{\text{old}}}(s,a) + \lambda\phi(\pi_{\text{new}}(a|s))]$$

Let $\tau_t = (s_0, a_0, \cdots, s_t, a_t)$ denotes the trajectory and $\tau$ is the whole trajectory (with infinite horizon). $\tau \sim \pi_{\text{old}}$ means the trajectory is generated by $\pi_{\text{old}}$. It follows that

$$
\begin{aligned}
& Q_\lambda^{\pi_{\text{old}}}(s_0, a_0) \\
&= \mathbb{E}[r(s_0, a_0) + \gamma \mathbb{E}_{s_1|\tau_0} V_\lambda^{\pi_{\text{old}}}(s_1)] \\
&= \mathbb{E}[r(s_0, a_0) + \gamma \mathbb{E}_{s_1|\tau_0} \mathbb{E}_{a_1 \sim \pi_{\text{old}}}[Q_\lambda^{\pi_{\text{old}}}(s_1, a_1) + \lambda\phi(\pi_{\text{old}}(a_1|s_1))]] \\
&\leq \mathbb{E}[r(s_0, a_0) + \gamma \mathbb{E}_{s_1|\tau_0} \mathbb{E}_{a_1 \sim \pi_{\text{new}}}[Q_\lambda^{\pi_{\text{old}}}(s_1, a_1) + \lambda\phi(\pi_{\text{new}}(a_1|s_1))]] \\
&= \mathbb{E}_{\tau_1 \sim \pi_{\text{new}}}[r(s_0, a_0) + \gamma(r(s_1, a_1) + \lambda\phi(\pi_{\text{new}}(a_1|s_1))) \quad + \gamma^2 \mathbb{E}_{s_2|\tau_1} V_\lambda^{\pi_{\text{old}}}(s_2)] \\
&\leq \mathbb{E}_{\tau_n \sim \pi_{\text{new}}}[r(s_0, a_0) + \sum_{t=1}^n \gamma^t(r(s_t, a_t) + \lambda\phi(\pi_{\text{new}}(a_t|s_t))) + \gamma^{n+1} \mathbb{E}_{s_{n+1}|\tau_n} V_\lambda^{\pi_{\text{old}}}(s_{n+1})] \\
&\leq \mathbb{E}_{\tau \sim \pi_{\text{new}}}[r(s_0, a_0) + \sum_{t=1}^\infty \gamma^t(r(s_t, a_t) + \lambda\phi(\pi_{\text{new}}(a_t|s_t)))] \\
&= Q_\lambda^{\pi_{\text{new}}}(s_0, a_0),
\end{aligned}
$$

where the last inequality is because we repeatedly expanded $Q_\lambda^{\pi_{\text{old}}}$ on the RHS by applying (27) and $Q_\lambda^{\pi_{\text{old}}}$ is bounded by $\frac{R_{\max}}{1-\gamma}$. ∎

**Proof for Theorem 6** Let $\pi_i$ be the policy at iteration $i$ of RPI. By Lemma 7, the sequence $Q_\lambda^{\pi_i}$ is monotonically increasing. Since $Q_\lambda^{\pi_i}$ is bounded by $\frac{R_{\max}}{1-\gamma}$ for any policy $\pi_i$, therefore $Q_\lambda^{\pi_i}$ will converge to a limit, denoted by $Q_\lambda^{\lim}$. Let $\pi_{\lim} = \arg\max_\pi \mathbb{E}_{a \sim \pi(\cdot|s)}[Q_\lambda^{\lim}(s, a) + \lambda\phi(\pi(a|s))]$. It is obvious that $Q_\lambda^{\pi_{\lim}} = Q_\lambda^{\lim}$. We aim to prove $\pi_{\lim} = \pi_\lambda^*$. To that end, we only need to prove $Q_\lambda^* = Q_\lambda^{\lim}$. For one hand, $Q_\lambda^{\pi_{\lim}}(s, a) = \lim_{n \to \infty} Q_\lambda^{\pi_i}(s, a) \leq Q_\lambda^*(s, a) = Q_\lambda^{\pi_\lambda^*}(s, a)$. For another hand, at convergence, it must be the case that for all policy $\pi$,

$$
\mathbb{E}_{a \sim \pi}[Q_\lambda^{\pi_{\lim}}(s, a) + \lambda\phi(\pi(a|s))] \leq \mathbb{E}_{a \sim \pi_{\lim}}[Q_\lambda^{\pi_{\lim}}(s, a) + \lambda\phi(\pi_{\lim}(a|s))].
$$

Using the same iterative argument as in the proof of Lemma 7, we get $Q_\lambda^{\pi_\lambda^*}(s, a) \leq Q_\lambda^{\pi_{\lim}}(s, a)$ for all states and actions. Putting above results together, it follows that $Q_\lambda^* = Q_\lambda^{\lim}$ therefore $\pi_{\lim} = \pi_\lambda^*$. ∎

# H   Experiment Details

## H.1   Discrete Environments

### H.1.1   Environment setup

For the random MDP model, we choose $|\mathcal{A}| = 50$, $|\mathcal{S}| = 10$ and $\gamma = 0.99$. Each state is assigned an index ranging from 0 to 49. The transition probabilites are generated by uniform distribution $[0, 1]$ and each entry of transition is clipped as zero with probability 0.95. Then each row of the clipped matrix is scaled to a probability distribution. The state we monitored is the state with index zero. The rewards are generated by uniform distribution $[0, 1]$. The initial Q-value is generated by uniform distribution $[0, 10]$ and policies are calculated explicitly or implicitly from Q-values.

For $(2N - 1) \times (2N - 1)$ GridWorld model, we choose $N = 10$ and $\gamma = 0.99$. The action space includes four actions (left, right, up, down). Each grid is indexed by an Cartesian coordinates $(x, y)$ with $x$ the row index and $y$ the column index. $x$ and $y$ are all range from $-(N - 1)$ to $N - 1$. The agent is initialized at the origin $(0, 0)$. Once it achieves four corners (i.e., $\pm(N - 1) \times \pm(N - 1)$), a reward with value 1 will be obtained. Otherwise, no reward will be given. Due to the symmetry of GridWorld, we are interesting on the three states $(0, 0), (0, N/2), (N/2, N/2)$. In the origin $(0, 0)$, all actions should be equal. While the agent locates at $(0, N/2)$ or $(N/2, N/2)$, the optimal policy should put more probability mass on the action which could lead to

### H.1.2 Optimization

In this section, we detail how we conduct RPI(Appendix F) in two discrete environments. Given a regularization function, we run 500 iterations of RPI that alternates between policy evaluation and policy improvement.

**Policy evaluation** Since in our experiments the transition probability is known, the evaluation of a given policy is conducted by DP. Specifically, let $\mathbb{P}^\pi \in \mathbb{R}^{|\mathcal{S}| \times |\mathcal{S}|}$ denote the transition matrix deduced from $\pi$, i.e., $\mathbb{P}^\pi(s, s') = \sum_{a'} \pi(a'|s)\mathbb{P}(s'|a', s)$ and $r_\lambda^\pi \in \mathbb{R}^{|\mathcal{S}|}$ the reward vector deduced from $\pi$, i.e., $r_\lambda^\pi(s) = \sum_{a'} r(s, a')\pi(a'|s)$. Then the regularized state value function $V_\lambda^\pi$ is solved from

$$V_\lambda^\pi = r_\lambda^\pi + \gamma \mathbb{P}^\pi V_\lambda^\pi \quad \Rightarrow \quad V_\lambda^\pi = (1 - \gamma \mathbb{P}^\pi)^{-1} r_\lambda^\pi$$

where by slightly notation abuse, $V_\lambda^\pi \in \mathbb{R}^{|\mathcal{S}|}$ is the vector with each coordinate $V_\lambda^\pi(s)$. Then $Q_\lambda^\pi$ can be computed from $V_\lambda^\pi$ by definition (4).

**Policy improvement** The policy improvement step involves an possibly intricate convex optimization (28). Here we detail how we solve the involved convex optimization.

Let $Q_\lambda^{\pi_{\mathrm{old}}}$ denote the already evaluated Q-value function of $\pi_{\mathrm{old}}$. For $\phi(x) = \frac{1}{2}(1 - x)$, since the improved policy has an explicit form [19]. However, for $\phi(x) = \cos(\frac{\pi}{2}x)$ and $\phi(x) = \exp(1) - \exp(x)$ which do not have an closed form and their corresponding $g_\phi$ are hard to formulate, thus we solve the convex optimization problem (28) directly. Specifically, for each $s \in \mathcal{S}$, we solve

$$\max_\pi \sum_a \pi(a|s) Q_\lambda^{\pi_{\mathrm{old}}}(s, a) + \lambda \sum_a \pi(a|s)\phi(\pi(a|s)).$$

In practice, we use CVXOPT [41] package to compute the improved policy.

### H.1.3 Regularizers

We test four *basic* regularizers, including $-\log x$, $\frac{1}{2}(1 - x)$, $\cos(\frac{\pi}{2}x)$ and $\exp(1) - \exp(x)$. From Proposition 1 and 2, we can combine different basic regularizers to more complicated ones, which we term as *combined* regularizers. We test the following three combined regularizers, (1) min: the minimum of tsallis and shannon, i.e., $\min\{-\log(x), 2(1 - x)\}$, (2) poly: the positive addition of two polynomial functions, i.e., $\frac{1}{2}(1 - x) + (1 - x^2)$ and (3) mix: the positive addition of tsallis and shannon, i.e., $-\log(x) + \frac{1}{2}(1 - x)$. We draw these seven regularizers and their corresponding $f'_\phi$ respectively in Figure 4 (a) and (b).

| (a) Different $\phi$'s | (b) $f'_\phi$ for different $\phi$'s | (c) Random MDP | (d) Gridworld |

Figure 4: (a) plots seven different regularization forms we will investigate. (b) shows the plot of $f'_\phi = \phi + x\phi'$ for corresponding regularizers. We prefer finite $f'_\phi(0)$ since it implies the optimal policy has a potentially sparse distribution if $\lambda$ is appropriately selected. (c) and (d) shows the results of the sparsity $\delta$ of the optimal policy on two envoirnments (Random MDP and Gridworld).

### H.1.4 Results for Random MDP

Figure 5 shows the probability mass of all actions in the optimal policy at selected state. When $\lambda$ is small, all regularizers except shannon have some zero-probability actions. When $\lambda$ is just over 2, exp and tsallis already have a full action support set. By contrast, cos is still sparse enough, implying the trigonometric function cos has a stronger ability in modeling sparseness. In the extreme case where $\lambda$ is sufficiently large, the optimal policy will converge to a uniform distribution on the action space as we expect.

(a) cos: $\cos(\frac{\pi}{2}x)$    (b) exp: $\exp(1) - \exp(x)$    (c) tsallis: $\frac{1}{2}(1-x)$    (d) shannon: $-\log x$

(e)
min:$\min\{-\log(x), 2(1 - x)\}$    (f) poly: $\frac{1}{2}(1-x)+(1-x^2)$ (g) mix: $-\log(x)+\frac{1}{2}(1-x)$

Figure 5: (a)-(g) shows the changing process of the probability mass on each action in the optimal policy in a random MDP where $|\mathcal{A}| = 10$. There are totally ten colored curves in each figure with one color representing one action.

### H.1.5    Results for Gridworld

Figure 6 shows the probability mass of four actions in the optimal policy at selected three states. When $\lambda$ is large, the optimal policies tend to uniform distribution. We show the result of three combined regularizars in Figure 7. It can be seen from these figures that in the regime of low $\lambda$, the optimal policy at different states show different preferrence. As shown in Random MDP, cos still has the strongest sparseness power.

### H.2    Atari Environments

We test our regularizers on OpenAI Gym benchmark with Atari environments: AlienNoFrameskip-v4, BoxingNoFrameskip-v4, BreakoutNoFrameskip-v4 and SeaquestNoFrameskip-v4.

**Architecture.**    We model the Q-values and policies with deep neural networks. The Q-value network is composed of 3 convolutional layers, 1 fully connected laryer, and 1 output fully connected layer as the following scheme. In particalr, the first convolutional layer $C_1$ has 32 $8 \times 8$ filter with stride 4, the second $C_2$ contains 64 $4 \times 4$ filters with stride 2, and the third $C_3$ has 64 $3 \times 3$ filters with stride 1. The fully connected layer $F_1$ consists of 512 hidden units and the layer $F_2$ is a $512 \times |\mathcal{A}|$ fully connected layer. Each layer except the final layer is followed with a rectified linear activation(ReLU). For shannon, the architecture of policy network is the same as Q-value network except the final layer is replaced with softmax function. For other regularizers with $0 \notin \mathrm{dom} f'_\phi$, the final layers are replaced with a softmax fully connected layer and a ReLU fully connected layer. The final ReLU fully connected layer serves as dual variables. The output probability is the multiplication of the two layers and scale the sum to 1. We use the Adam optimizer with learning rate 0.0001 and $\varepsilon = 0.0015$. The discount was set to $\gamma = 0.99$. We update the target network every 10000 steps. The size of experience replay buffer is 100000 tuples, where 32 minibatches were sampled every 4 steps to update the network.

Q-value : $C_1 \xrightarrow{ReLU} C_2 \xrightarrow{ReLU} C_3 \xrightarrow{ReLU} F_1 \xrightarrow{ReLU} F_2$

Policy (shannon) : $C_1 \xrightarrow{ReLU} C_2 \xrightarrow{ReLU} C_3 \xrightarrow{ReLU} F_1 \xrightarrow{Softmax} F_2$

Figure 6: The probability mass on four actions in the optimal policy regularized by four basic regularization functions at selected three states. (a)-(d) shows the results for the origin $(0,0)$. (e)-(h) shows the results for the state $(0, N/2)$ and (i)-(l) shows the results for the state $(N/2, N/2)$

$$\text{Policy (Other)} : C_1 \xrightarrow{ReLU} C_2 \xrightarrow{ReLU} C_3 \xrightarrow{ReLU} F_1 \xrightarrow{Softmax} F_2 \longrightarrow F_4 \xrightarrow{Scale} F_5$$

$$F_1 \xrightarrow{ReLU} F_3 \xrightarrow{\odot} F_4$$

**Parameter sensitivity.** We show how learning performance changes when $\lambda$ varies in Figure 8. Large $\lambda$ will make the policy becomes nearly uniform and unable to make use of the information of rewards. Small $\lambda$ will make the policy becomes nearly deterministic and therefore be stuck in local minima since no sufficient exploration is made. In the experiment of Breakout, we find that `shannon` is insensitive to $\lambda$. However, for other regularizers, small or large $\lambda$ would make the algorithm fail to converge.

## H.3 Mujoco Environments

We choose OpenAI Gym benchmark with the MuJoCo simulator for our test environments, including Hopper-v2, Walker-v2, HalfCheetah-v2 and Ant-v2. We exclude `cos` due to its numerical unstability. Then we only consider three regularizers, i.e., $-\log x$, $\frac{1}{2}(1-x)$ and $\exp(1) - \exp(x)$ for their stable performance in deep RL training process. Since RAC is very similar to SAC except RAC is agonostic to regularization forms. We build our code on the work of SAC [14]. For comparison's purpose, we use the same network structure and hyper-parameter settings. Figure 9 shows that the full experiments we conducted in each environment. Each regularizer is coupled with three regularization parameter $\lambda \in \{1, 0.1, 0.01\}$.

**The reparameterization trick.** Mujoco is continuous problem, we model the policy $\pi_\psi(a|s)$ as a factorized Gaussian distribution with the mean and variance modeled as neural networks. Besides, we can update policy parameters like eqn(10) as there is no access to compute expectation over $\pi$

(a) `min`: $\min\{-\log(x), 2(1-x)\}$  (b) `poly`: $\frac{1}{2}(1-x)+(1-x^2)$  (c) `mix`: $-\log(x)+\frac{1}{2}(1-x)$

(d) `min`: $\min\{-\log(x), 2(1-x)\}$  (e) `poly`: $\frac{1}{2}(1-x)+(1-x^2)$  (f) `mix`: $-\log(x)+\frac{1}{2}(1-x)$

(g) `min`: $\min\{-\log(x), 2(1-x)\}$  (h) `poly`: $\frac{1}{2}(1-x)+(1-x^2)$  (i) `mix`: $-\log(x)+\frac{1}{2}(1-x)$

Figure 7: The probability mass on four actions in the optimal policy regularized by three combined regularization functions at selected three states. (a)-(c) shows the results for the origin $(0,0)$. (d)-(f) shows the results for the state $(0, N/2)$ and (g)-(h) shows the results for the state $(N/2, N/2)$.

(a) Breakout Shannon    (b) Breakout Tsallis    (c) Breakout Expx    (d) Breakout Cosx

Figure 8: Training curves on Atari games. Each entry in the legend is named with the rule `the regularization form` $+ \lambda$. The score is smoothed with 100 windows while the shaded area is the one standard deviation.

in continuous setting. We apply the reparameterization trick to update the policy. The policy is reparameterized as an factorized gaussian with tanh output, i.e.,

$$\pi_\psi(s, \epsilon) = \tanh(\text{mean}_\psi(s) + \epsilon \cdot \text{std}_\psi(s))$$

where $\epsilon$ is an input noise vector, sampled from the standard Gaussian. Denote the generative action $a_t = \tanh(Z_t)$ and $Z_t$ is a multivariate normal distribution, we have the density transformation $\pi(a_t) = \mathcal{N}(Z_t)|\det(\frac{da_t}{dZ_t})|^{-1}$, where $\log\det(\frac{da_t}{dZ_t}) = \sum_{i=1}^{\mathcal{A}}\log(1-\tanh^2(Z_{t,i}))$. Therefore, we can rewrite the policy loss as:

$$J_\pi(\psi) = \hat{\mathbb{E}}_\mathcal{D}\left[-\lambda\phi(\pi_\psi(s_t, \epsilon_t)) - Q_\theta(s_t, \phi(\pi_\psi(s_t, \epsilon_t)))\right]. \tag{29}$$

We now approximate the gradient of $J_\pi(\psi)$ with:

$$\hat{\nabla} J_\pi(\psi) = \nabla_\psi \phi(\pi_\psi(a_t|s_t)) + (\nabla_{a_t}\phi(\pi_\psi(a_t|s_t)) - \nabla_{a_t} Q(s_t, a_t))\nabla_\psi \pi_\psi(s_t, \epsilon_t),$$

where $a_t$ is evaluated at $\pi_\psi(s_t, \epsilon_t)$.

**Parameter sensitivity.** As reported by Haarnoja et al. [13], shannon is very sensitive to the regularization coefficient $\lambda$ (which is also referred as the temperature parameter). As an extreme example, when $\lambda = 1$, shannon fails to converge in Walker-v2 and Ant-v2. By contrast, tsallis is less sensitive to $\lambda$. As $\lambda$ varies from 0.01 to 1, the performance of tsallis doesn't degrade to much. Exp is also insensitive to hyperparameter $\lambda$.

(a) Ant-v2          (b) Walker-v2          (c) Hopper-v2          (d) HalfCheetah-v2

Figure 9: Training curves on continuous control benchmarks. Each curve is the average of four experiments with different seeds. Each entry in the legend is named with the rule the regularization form $+ \lambda$. The score is smoothed with 30 windows while the shaded area is the one standard deviation.