[Reviews · NeurIPS 2019]

Reviewer 1



Originality: There is a lot of papers studying regularized RL problems (cited by this paper). Regularization can introduce many good properties to the solution of the MDPs. This paper proposes a generalized framework that includes most of the previously studied regularization based methods. This framework can lead to effective understanding of which regularizer is sparsity-generating. To my knowledge, I believe this paper contains sufficient originality. Quality: I did not check the proof line by line. But it appears to be correct. The experimental results also make sense. The study is quite complete. Clarity: This paper is well-written and is of publishing quality. Significance: The paper gives a good study of the regularized framework of solving MDPs. It provides some interesting understanding of the regularizers. I believe it is sufficiently significant to be accepted. Minor comments: * What does it mean by "multi-modality"?

Reviewer 2



-Originality: The characterization of optimal policy and bounds on the value functions are new. Related work is cited. Although some techniques are analogous to previous work (which is not bad per se, as it allows to apply more general regularisers within previous frameworks such as soft-actor-critic with small changes only), this work differs significantly from previous work and yields new insights how to obtain sparse policies or not. -Quality: The submission seems largely technically sound. Claims are supported by proofs and experiments confirm that considering more flexible regularizations can be beneficial in different tasks. There are some issues with the continuous time case, see the section on improvements for details. Further the authors claim that trigonometric and exponential functions families yield multimodal policies (line 287). However, it is not clear to me how this is different to say entropy regularisation, and why a softmax policy cannot have multiple modes (unless of course I parameterize the policy with a single Gaussian in the continuous case, but this is a different issue). Also I am not sure if I understand line 123-124, it seems to me contradictory to Proposition 1 that \mathcal{F} is closed under min. Lastly, I was also wondering how fast and scalable this approach with solving the convex optimization problem via CVXOPT is versus gradient descent approaches used in entropy-regularised settings with function approximators? -Clarity: The paper is relatively well written and organized. The convergence of a regularised policy iteration algorithm (Theorem 6) is just given in the appendix, maybe it is useful to mention this somehow in the main text also. -Significance: It is important to have a better understanding of how different regularisers affect the sparsity of the optimal policy and resulting performance errors. The contribution addresses these issues and I feel that others can build upon and benefit from the ideas presented therein. POST AUTHOR RESPONSE: I thank the authors for their feedback. Having read it along with the other reviews, I increase my score from 6 to 7. The rebuttal has largely addressed my concerns in the continuous-state case. The missing log-determinant term in the density seems to be more a typo and has apparently been included in the experiments. I am not convinced that the chosen paramterisation for the density is a particularly good choice to obtain sparse policies. However, it is a commonly used transformation (such as in entropy-regularised approaches), so I think that this choice is actually ok, also as the continuous-action case is not the main focus of the paper. The response also commented on the scalability of the proposed optimization approach versus stochastic gradient descent commonly used in related work. Overall the paper is interesting and can be accepted.

Reviewer 3



This paper is incredibly thorough and represents a clear advancement to our understanding of regularization in MDPs. (See the contributions listed above.)

[Author Response · NeurIPS 2019]

We greatly appreciate the reviewers' effort and helpful comments. We are improving the paper by incorporating the
reviewers' suggestions.

**R#1** 1. A policy has multi-modality means it assigns positive probability mass to optimal and near optimal actions.
This concept is more useful when the multiple optimal (or near optimal) actions exist. In this case, a policy with
multi-modality is a distribution over these (near) optimal actions and provides information of multiple actions (including
the probability weight) in a state, unlike the deterministic one which provides only one action information.

2. Based on our results, among the four basic regularizers, cos has the strongest sparsity power (Lines 257-259). This
regularizer performs better in scenarios with high-dimensional action space. For example, the performance of cos
increases fastest in the environment of Seaquest where $|\mathcal{A}| = 18$ (see (d) in Figure 2). Thanks!

**R#2** 1. Line 734: Yes, it's Tanh transformation. The random variable is transformed as $a_t = \tanh(Z_t)$, and the
corresponding density relation is $\pi(a_t|s_t) = \mathcal{N}(Z_t)|\det(\frac{da_t}{dZ_t})|^{-1}$, where $\log|\det(\frac{da_t}{dZ_t})| = \sum_{i=1}^{\mathcal{A}} \log(1-\tanh^2(Z_{t,i}))$.
We have taken it into account but miss it in the appendix. We will add up the details in its revision.

2. Tanh sensible choice?: (i) Indeed, the similar theoretical results can be derived in continuous action space by
variational calculus. As $\lambda$ goes zero, the density function reduces to Dirac function centered at $\arg\max_a Q^*(s,a)$.
Besides, the density reduces to zero exactly for $a \notin \arg\max_a Q^*(s,a)$ when sparse regularizers are applied. But, in
empirical continuous action space setting, it is difficult to find a tractable way to model a function class containing
Dirac function and density functions that can be sparse. (You mention that some prior densities can be used to induce
sparsity, which is a good advice. We will consider it in the future.) So we just follow the Mujoco experiment setting in
(Haarnoja et al. "Soft actor-critic: Off-policy maximum entropy deep reinforcement learning with a stochastic actor." )
to check whether the algorithm still performs well with our proposed regularizers.

(ii) In discrete action space, we require the trigonometric family functions are non-negative as the usual entropy (e.g.
Shannon entropy), which is lower bounded by 0. The added non-negative term can be viewed as a bonus.

3. Line 287: These regularizers have multi-modality (which we have explained in R#1.1) as soft-max policy does. But
soft-max policy doesn't have sparsity since it assigns positive probabilities to all actions.

4. Lines 123-124: The only consideration is whether the regularizer is differentiable. For ease of analysis, we only
consider the case where $\phi(x)$ is fully differentiable. Actually, we can relax (3) in Assumption 1 such that $\phi(x)$ is
differential except for finite points. And our theoretical results still hold in this case.

5. CVXOPT or GD: It depends on the size of problem. If the problem is simple (small state and action space), CVXOPT
converges faster. But if the problem is complicated (e.g., Atari) with large samples, CVXOPT is slow as we have to
solve a convex problem for each sample $(s,a,s',r)$ in a batch, which is time-consuming when $|\mathcal{A}|, |\mathcal{S}|$ are large. GD is
more scalable by contrast. Thanks!

**R#3** Thanks for your careful review. We will incorporate your comments and suggestions into the revision.

1. Line 17: Sorry, our phrasing is not careful. Here our point is that greedily solving the Bellman equation is not an easy
task in a high-dimension action space or when function approximation (such as neural networks) is used.

2. Lines 18-19: Yes, it is possible to break ties randomly. But our point is to emphasize the need of information of
multiple actions. Multiple (near) optimal actions provide alternatives when the suggested action is suddenly forbidden.

3. Line 21: If the task is path planning where the state is the pair of departure and destination, when the suggested routine
is unfortunately congested, an alternative routine could be provided by a multi-modal policy. In this way, we don't need
to evoke the computation of new routines. The example aims to shed light on the importance of multi-modality.

4. Line 24: Yes, we want the learning problem is computationally inexpensive and the optimal policy has multi-modality.

5. Line 27: Yes, it's unclear. We will delete it.

6. Line 31-32: Yes, we don't want terrible actions have a chance to be executed. We will modify this sentence.

7. Line 49-50: The point is that there exist other regularizers that induces sparsity.

8. Line 70: It's a typo. We require $\gamma$ to be non-negative in our proof and our theories still hold when $\gamma = 0$.

9. Lines 102-103: Yes, I agree with you. We will add this distinction in the revision.

10. Theorem 1: The primal problem (Eqn. 2) can be transformed into a convex problem about the vector $\{\pi(a|s)\}_{s,a}$,
so Theorem 1 is necessary and sufficient as it is derived from the KKT conditions.

[Meta-Review · NeurIPS 2019]

The paper contributes valuable insights to understanding the influence of regularization in policy optimization in MDPs. The reviewers reached a consensus about the paper's results having significant merit, and the paper can be accepted for presentation.